# Ctrl-X: Controlling Structure and Appearance for Text-To-Image Generation Without Guidance

**Kuan Heng Lin**[1*] **Sicheng Mo**[1*] **Ben Klingher**[1] **Fangzhou Mu**[2] **Bolei Zhou**[1]
[1]University of California, Los Angeles [2]NVIDIA

https://genforce.github.io/ctrl-x/

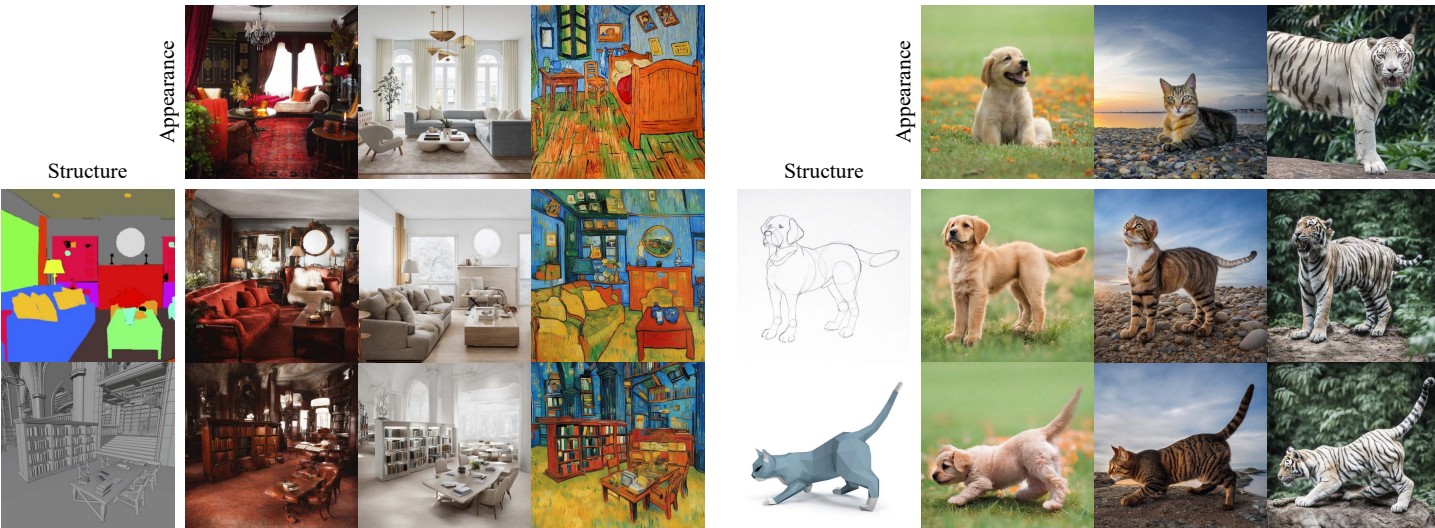

Figure 1: **Guidance-free structure and appearance control of Stable Diffusion XL (SDXL) [27]** Ctrl-X enables training-free and guidance-free zero-shot control of pretrained text-to-image diffusion models given any structure conditions and appearance images.

## Abstract

Recent controllable generation approaches such as FreeControl [24] and Diffusion Self-Guidance [7] bring fine-grained spatial and appearance control to text-to-image (T2I) diffusion models without training auxiliary modules. However, these methods optimize the latent embedding for each type of score function with longer diffusion steps, making the generation process time-consuming and limiting their flexibility and use. This work presents *Ctrl-X*, a simple framework for T2I diffusion controlling structure and appearance without additional training or guidance. Ctrl-X designs feed-forward structure control to enable the structure alignment with a structure image and semantic-aware appearance transfer to facilitate the appearance transfer from a user-input image. Extensive qualitative and quantitative experiments illustrate the superior performance of Ctrl-X on various condition inputs and model checkpoints. In particular, Ctrl-X supports novel structure and appearance control with arbitrary condition images of any modality, exhibits superior image quality and appearance transfer compared to existing works, and provides instant plug-and-play functionality to any T2I and text-to-video (T2V) diffusion model.

---

[*]Indicates equal contribution

38th Conference on Neural Information Processing Systems (NeurIPS 2024).

# 1 Introduction

The rapid advancement of large text-to-image (T2I) generative models has made it possible to generate high-quality images with just one text prompt. However, it remains challenging to specify the exact concepts that can accurately reflect human intents using only textual descriptions. Recent approaches like ControlNet [44] and IP-Adapter [43] have enabled controllable image generation upon pretrained T2I diffusion models regarding structure and appearance, respectively. Despite the impressive results in controllable generation, these approaches [44, 25, 46, 20] require fine-tuning the entire generative model or training auxiliary modules on large amounts of paired data.

Training-free approaches [7, 24, 4] have been proposed to address the high overhead associated with additional training stages. These methods optimize the latent embedding across diffusion steps using specially designed score functions to achieve finer-grained control than text alone with a process called guidance. Although training-free approaches avoid the training cost, they significantly increase computing time and required GPU memory in the inference stage due to the additional backpropagation over the diffusion network. They also require sampling steps that are 2–20 times longer. Furthermore, as the expected latent distribution of each time step is predefined for each diffusion model, it is critical to tune the guidance weight delicately for each score function; Otherwise, the latent might be out-of-distribution and lead to artifacts and reduced image quality.

To tackle these limitations, we present *Ctrl-X*, a simple *training-free* and *guidance-free* framework for T2I diffusion with structure and appearance control. We name our method "Ctrl-X" because we reformulate the controllable generation problem by 'cutting' (and 'pasting') two tasks together: spatial structure preservation and semantic-aware stylization. Our insight is that diffusion feature maps capture rich spatial structure and high-level appearance from early diffusion steps sufficient for structure and appearance control without guidance. To this end, Ctrl-X employs feature injection and spatially-aware normalization in the attention layers to facilitate structure and appearance alignment with user-provided images. By being guidance-free, Ctrl-X eliminates additional optimization overhead and sampling steps, resulting in a 35-fold increase in inference speed compared to guidance-based methods. Figure 1 shows sample generation results. Moreover, Ctrl-X supports arbitrary structure conditions beyond natural images and can be applied to any T2I and even text-to-video (T2V) diffusion models. Extensive quantitative and qualitative experiments, along with a user study, demonstrate the superior image quality and appearance alignment of our method over prior works.

We summarize our contributions as follows:

1. We present *Ctrl-X*, a simple plug-and-play method that builds on pretrained text-to-image diffusion models to provide disentangled and zero-shot control of structure and appearance during the generation process requiring no additional training or guidance.
2. Ctrl-X presents the first universal guidance-free solution that supports multiple conditional signals (structure and appearance) and model architectures (*e.g.* text-to-image and text-to-video).
3. Our method demonstrates superior results in comparison to previous training-based and guidance-based baselines (*e.g.* ControlNet + IP-Adapter [44, 43] and FreeControl [24]) in terms of condition alignment, text-image alignment, and image quality.

# 2 Related work

**Diffusion structure control.** Previous spatial structure control methods can be categorized into two types (training-based *vs.* training-free) based on whether they require training on paired data.

*Training-based structure control methods* require paired condition-image data to train additional modules or fine-tune the entire diffusion network to facilitate generation from spatial conditions [44, 25, 20, 46, 42, 3, 47, 38, 49]. While pixel-level spatial control can be achieved with this approach, a significant drawback is needing a large number of condition-image pairs as training data. Although some condition data can be generated from pretrained annotators (*e.g.* depth and segmentation maps), other condition data is difficult to obtain from given images (*e.g.* 3D mesh, point cloud), making these conditions challenging to follow. Compared to these training-based methods, Ctrl-X supports conditions where paired data is challenging to obtain, making it a more flexible and effective solution.

*Training-free structure control methods* typically focus on specific conditions. For example, R&B [40] facilitates bounding-box guided control with region-aware guidance, and DenseDiffusion [17] gen-

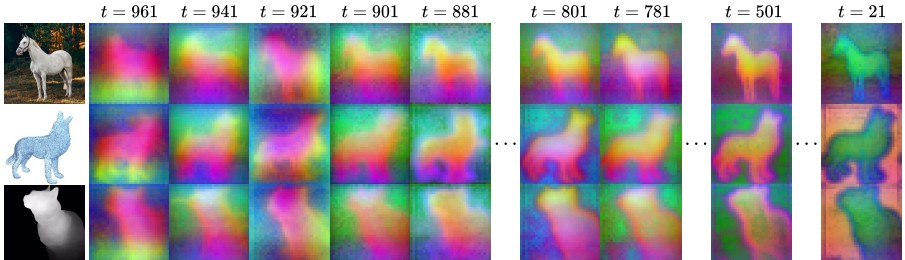

Figure 2: **Visualizing early diffusion features.** Using 20 real, generated, and condition images of animals, we extract Stable Diffusion XL [27] features right after decoder layer 0 convolution. We visualize the top three principal components computed for each time step across all images. $t = 961$ to $881$ correspond to inference steps 1 to 5 of the DDIM scheduler with 50 time steps. We obtain $\mathbf{x}_t$ by directly adding Gaussian noise to each clean image $\mathbf{x}_0$ via the diffusion forward process.

erates images with sparse segmentation map conditions by manipulating the attention weights. Universal Guidance [4] employs various pretrained classifiers to support multiple types of condition signals. FreeControl [24] analyzes semantic correspondence in the subspace of diffusion features and harnesses it to support spatial control from any visual condition. While these approaches do not require training data, they usually need to compute the gradient of the latent to lower an auxiliary loss, which requires substantial computing time and GPU memory. In contrast, Ctrl-X requires no guidance at the inference stage and controls structure via direct feature injections, enabling faster and more robust image generation with spatial control.

**Diffusion appearance control.** Existing appearance control methods that build upon pretrained diffusion models can also similarly be categorized into two types (training-based *vs.* training-free).

*Training-based appearance control methods* can be divided into two categories: Those trained to handle any image prompt and those overfitting to a single instance. The first category [44, 25, 43, 38] trains additional image encoders or adapters to align the generated process with the structure or appearance from the reference image. The second category [30, 14, 8, 2, 26, 31] is typically applied to customized visual content creation by finetuning a pretrained text-to-image model on a small set of images or binding special tokens to each instance. The main limitation of these methods is that the additional training required makes them unscalable. However, Ctrl-X offers a scalable solution to transfer appearance from any instance without training data.

*Training-free appearance control methods* generally follow two approaches: One approach [1, 5, 41] manipulates self-attention features using pixel-level dense correspondence between the generated image and the target appearance, and the other [7, 24] extracts appearance embeddings from the diffusion network and transfers the appearance by guiding the diffusion process towards the target appearance embedding. A key limitation of these approaches is that a single text-controlled target cannot fully capture the details of the target image, and the latter methods require additional optimization steps. By contrast, our method exploits the spatial correspondence of self-attention layers to achieve semantically-aware appearance transfer without targeting specific subjects.

## 3 Preliminaries

Diffusion models are a family of probabilistic generative models characterized by two processes: The *forward process* iteratively adds Gaussian noise to a clean image $\mathbf{x}_0$ to obtain $\mathbf{x}_t$ for time step $t \sim [1, T]$, which can be reparameterized in terms of a noise schedule $\alpha_t$ where

$$\mathbf{x}_t = \sqrt{\alpha_t}\mathbf{x}_0 + \sqrt{1 - \alpha_t}\epsilon \tag{1}$$

for $\epsilon \sim \mathcal{N}(0, \mathbf{I})$; The *backward process* generates images by iteratively denoising an initial Gaussian noise $\mathbf{x}_T \sim \mathcal{N}(0, \mathbf{I})$, also known as diffusion sampling [13]. This process uses a parameterized denoising network $\epsilon_\theta$ conditioned on a text prompt $\mathbf{c}$, where at time step $t$ we obtain a cleaner $\mathbf{x}_{t-1}$

$$\mathbf{x}_{t-1} = \sqrt{\alpha_{t-1}}\hat{\mathbf{x}}_0 + \sqrt{1 - \alpha_{t-1}}\epsilon_\theta(\mathbf{x}_t \mid t, \mathbf{c}), \qquad \hat{\mathbf{x}}_0 := \frac{\mathbf{x}_t - \sqrt{1 - \alpha_t}\epsilon_\theta(\mathbf{x}_t \mid t, \mathbf{c})}{\sqrt{\alpha_t}}. \tag{2}$$

Formally, $\epsilon_\theta(\mathbf{x}_t \mid t, \mathbf{c}) \approx -\sigma_t\nabla_{\mathbf{x}}\log p_t(\mathbf{x}_t \mid t, \mathbf{c})$ approximates a score function scaled by a noise schedule $\sigma_t$ that points toward a high density of data, i.e., $\mathbf{x}_0$, at noise level $t$ [34].

**Guidance.** The iterative inference of diffusion enables us to guide the sampling process on auxiliary information. *Guidance* modifies Equation 2 to compose additional score functions that point toward richer and specifically conditioned distributions [4, 7], expressed as

$$\hat{\epsilon}_\theta(\mathbf{x}_t \mid t, \mathbf{c}) = \epsilon(\mathbf{x}_t \mid t, \mathbf{c}) - s\,\mathbf{g}(\mathbf{x}_t \mid t, y), \tag{3}$$

where $\mathbf{g}$ is an energy function and $s$ is the guidance strength. In practice, $\mathbf{g}$ can range from classifier-free guidance (where $\mathbf{g} = \epsilon$ and $y = \varnothing$, *i.e.* the empty prompt) to improve image quality and prompt adherence for T2I diffusion [12, 29], to arbitrary gradients $\nabla_{\mathbf{x}_t}\ell(\epsilon(\mathbf{x}_t \mid t, \mathbf{c}) \mid t, y)$ computed from auxiliary models or diffusion features common to guidance-based controllable generation [4, 7, 24]. Thus, guidance provides great customizability on the type and variety of conditioning for controllable generation, as it only requires any loss that can be backpropagated to $\mathbf{x}_t$. However, this backpropagation requirement often translates to slow inference time and high memory usage. Moreover, as guidance-based methods often compose multiple energy functions, tuning the guidance strength $s$ for each $\mathbf{g}$ may be finicky and cause issues of robustness. Thus, Ctrl-X avoids guidance and provides instant applicability to larger T2I and T2V models with minor hyperparameter tuning.

**Diffusion U-Net architecture.** Many pretrained T2I diffusion models are text-conditioned U-Nets, which contain an encoder and a decoder that downsample and then upsample the input $\mathbf{x}_t$ to predict $\epsilon$, with long skip connections between matching encoder and decoder resolutions [13, 29, 27]. Each encoder/decoder block contains convolution layers, self-attention layers, and cross-attention layers: The first two control both structure and appearance, and the last injects textual information. Thus, many training-free controllable generation methods utilize these layers, through direct manipulation [11, 36, 18, 1, 41] or for computing guidance losses [7, 24], with self-attention most commonly used: Let $\mathbf{h}_{l,t} \in \mathbb{R}^{(hw)\times c}$ be the diffusion feature with height $h$, width $w$, and channel size $c$ at time step $t$ right before attention layer $l$. Then, the self-attention operation is

$$\mathbf{Q} := \mathbf{h}_{l,t}\mathbf{W}_l^Q \quad \text{and} \quad \mathbf{K} := \mathbf{h}_{l,t}\mathbf{W}_l^K \quad \text{and} \quad \mathbf{V} := \mathbf{h}_{l,t}\mathbf{W}_l^V,$$

$$\mathbf{h}_{l,t} \leftarrow \mathbf{A}\mathbf{V}, \qquad \mathbf{A} := \mathrm{softmax}\left(\frac{\mathbf{Q}\mathbf{K}^\top}{\sqrt{d}}\right), \tag{4}$$

where $\mathbf{W}_l^Q, \mathbf{W}_l^K, \mathbf{W}_l^V \in \mathbb{R}^{c\times d}$ are linear transformations which produce the query $\mathbf{Q}$, key $\mathbf{K}$, and value $\mathbf{V}$, respectively, and $\mathrm{softmax}$ is applied across the second $(hw)$-dimension. (Generally, $c = d$ for diffusion models.) Intuitively, the attention map $\mathbf{A} \in \mathbb{R}^{(hw)\times(hw)}$ encodes how each pixel in $\mathbf{Q}$ corresponds to each in $\mathbf{K}$, which then rearranges and weighs $\mathbf{V}$. This correspondence is the basis for Ctrl-X's spatially-aware appearance transfer.

# 4 Guidance-free structure and appearance control

Ctrl-X is a general framework for training-free, guidance-free, and zero-shot T2I diffusion with structure and appearance control. Given a structure image $\mathbf{I}^s$ and appearance image $\mathbf{I}^a$, Ctrl-X manipulates a pretrained T2I diffusion model $\epsilon_\theta$ to generate an output image $\mathbf{I}^o$ that inherits the structure of $\mathbf{I}^s$ and appearance of $\mathbf{I}^a$.

**Method overview.** Our method is illustrated in Figure 3 and is summarized as follows: Given clean structure and appearance latents $\mathbf{I}^s = \mathbf{x}_0^s$ and $\mathbf{I}^a = \mathbf{x}_0^a$, we first directly obtain noised structure and appearance latents $\mathbf{x}_t^s$ and $\mathbf{x}_t^a$ via the diffusion forward process, then extract their U-Net features from a pretrained T2I diffusion model. When denoising the output latent $\mathbf{x}_t^o$, we inject convolution and self-attention features from $\mathbf{x}_t^s$ and leverage self-attention correspondence to transfer spatially-aware appearance statistics from $\mathbf{x}_t^a$ to $\mathbf{x}_t^o$ to achieve structure and appearance control.

## 4.1 Feed-forward structure control

Structure control of T2I diffusion requires transferring structure information from $\mathbf{I}^s = \mathbf{x}_0^s$ to $\mathbf{x}_t^o$, especially during early time steps. To this end, we initialize $\mathbf{x}_T^o = \mathbf{x}_T^s \sim \mathcal{N}(0, \mathbf{I})$ and obtain $\mathbf{x}_t^s$ via the diffusion forward process in Equation 1 with $\mathbf{x}_0^s$ and randomly sampled $\epsilon \sim \mathcal{N}(0, \mathbf{I})$. Inspired by the observation where diffusion features contain rich layout information [36, 18, 24], we perform feature and self-attention injection as follows: For U-Net layer $l$ and diffusion time step $t$, let $\mathbf{f}_{l,t}^o$ and $\mathbf{f}_{l,t}^s$ be features/activations after the convolution block from $\mathbf{x}_t^o$ and $\mathbf{x}_t^s$, and let $\mathbf{A}_{l,t}^o$ and $\mathbf{A}_{l,t}^s$ be the attention maps of the self-attention block from $\mathbf{x}_t^o$ and $\mathbf{x}_t^s$. Then, we replace

$$\mathbf{f}_{l,t}^o \leftarrow \mathbf{f}_{l,t}^s \quad \text{and} \quad \mathbf{A}_{l,t}^o \leftarrow \mathbf{A}_{l,t}^s. \tag{5}$$

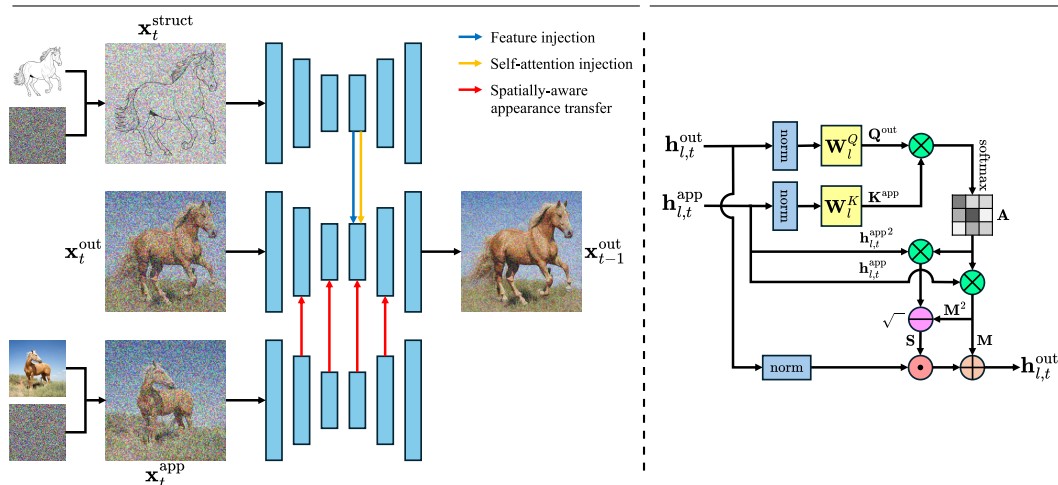

(a) Ctrl-X pipeline    (b) Spatially-aware appearance transfer

Figure 3: **Overview of Ctrl-X.** (a) At each sampling step $t$, we obtain $\mathbf{x}_t^{\mathrm{s}}$ and $\mathbf{x}_t^{\mathrm{a}}$ via the forward diffusion process, then feed them into the T2I diffusion model to obtain their convolution and self-attention features. Then, we inject convolution and self-attention features from $\mathbf{x}_t^{\mathrm{s}}$ and leverage self-attention correspondence to transfer spatially-aware appearance statistics from $\mathbf{x}_t^{\mathrm{a}}$ to $\mathbf{x}_t^{\mathrm{o}}$. (b) Details of our spatially-aware appearance transfer, where we exploit self-attention correspondence between $\mathbf{x}_t^{\mathrm{o}}$ and $\mathbf{x}_t^{\mathrm{a}}$ to compute weighted feature statistics $\mathbf{M}$ and $\mathbf{S}$ applied to $\mathbf{x}_t^{\mathrm{o}}$.

In contrast to [36, 18, 24], we do not perform inversion and instead directly use forward diffusion (Equation 1) to obtain $\mathbf{x}_t^{\mathrm{s}}$. We observe that $\mathbf{x}_t^{\mathrm{s}}$ obtained via the forward diffusion process contains sufficient structure information even at *very* early/high time steps, as shown in Figure 2. This also reduces appearance leakage common to inversion-based methods observed by FreeControl [24]. We study our feed-forward structure control method in Sections 5.1 and 5.2.

We apply feature injection for layers $l \in L^{\mathrm{feat}}$ and self-attention injection for layers $l \in L^{\mathrm{self}}$, and we do so for (normalized) time steps $t \leq \tau^{\mathrm{s}}$, where $\tau^{\mathrm{s}} \in [0, 1]$ is the structure control schedule.

## 4.2 Spatially-aware appearance transfer

Inspired by prior works that define appearance as feature statistics [15, 21], we consider appearance transfer to be a stylization task. T2I diffusion self-attention transforms the value $\mathbf{V}$ with attention map $\mathbf{A}$, where the latter represents how pixels in $\mathbf{Q}$ corresponds to pixels in $\mathbf{K}$. As observed by Cross-Image Attention [1], $\mathbf{Q}\mathbf{K}^\top$ can represent the semantic correspondence between two images when $\mathbf{Q}$ and $\mathbf{K}$ are computed from features from each, even when the two images differ significantly in structure. Thus, inspired by AdaAttN [21], we propose spatially-aware appearance transfer, where we exploit this correspondence to generate self-attention-weighted mean and standard deviation maps from $\mathbf{x}_t^{\mathrm{a}}$ to normalize $\mathbf{x}_t^{\mathrm{o}}$: For any self-attention layer $l$, let $\mathbf{h}_{l,t}^{\mathrm{o}}$ and $\mathbf{h}_{l,t}^{\mathrm{a}}$ be diffusion features right before self-attention for $\mathbf{x}_t^{\mathrm{o}}$ and $\mathbf{x}_t^{\mathrm{a}}$, respectively. Then, we compute the attention map

$$\mathbf{A} = \mathrm{softmax}\left(\frac{\mathbf{Q}^{\mathrm{o}}\mathbf{K}^{\mathrm{a}\top}}{\sqrt{d}}\right), \qquad \mathbf{Q}^{\mathrm{o}} := \mathrm{norm}(\mathbf{h}_{l,t}^{\mathrm{o}})\mathbf{W}_l^Q \quad \text{and} \quad \mathbf{K}^{\mathrm{a}} := \mathrm{norm}(\mathbf{h}_{l,t}^{\mathrm{a}})\mathbf{W}_l^K, \quad (6)$$

where $\mathrm{norm}$ is applied across spatial dimension $(hw)$. Notably, we normalize $\mathbf{h}_{l,t}^{\mathrm{o}}$ and $\mathbf{h}_{l,t}^{\mathrm{a}}$ first to remove appearance statistics and thus isolate structural correspondence. Then, we compute the mean and standard deviation maps $\mathbf{M}$ and $\mathbf{S}$ of $\mathbf{h}_{l,t}^{\mathrm{a}}$ weighted by $\mathbf{A}$ and use them to normalize $\mathbf{h}_{l,t}^{\mathrm{o}}$,

$$\mathbf{h}_{l,t}^{\mathrm{o}} \leftarrow \mathbf{S} \odot \mathbf{h}_{l,t}^{\mathrm{o}} + \mathbf{M}, \qquad \mathbf{M} := \mathbf{A}\mathbf{h}_{l,t}^{\mathrm{a}} \quad \text{and} \quad \mathbf{S} := \sqrt{\mathbf{A}(\mathbf{h}_{l,t}^{\mathrm{a}} \odot \mathbf{h}_{l,t}^{\mathrm{a}}) - (\mathbf{M} \odot \mathbf{M})}. \quad (7)$$

$\mathbf{M}$ and $\mathbf{S}$, weighted by structural correspondences between $\mathbf{I}^{\mathrm{o}}$ and $\mathbf{I}^{\mathrm{a}}$, are spatially-aware feature statistics of $\mathbf{x}_t^{\mathrm{a}}$ which are transferred to $\mathbf{x}_t^{\mathrm{o}}$. Lastly, we perform layer $l$ self-attention on $\mathbf{h}_{l,t}^{\mathrm{o}}$ as normal.

We apply appearance transfer for layers $l \in L^{\mathrm{app}}$, and we do so for (normalized) time steps $t \leq \tau^{\mathrm{a}}$, where $\tau^{\mathrm{a}} \in [0, 1]$ is the appearance control schedule.

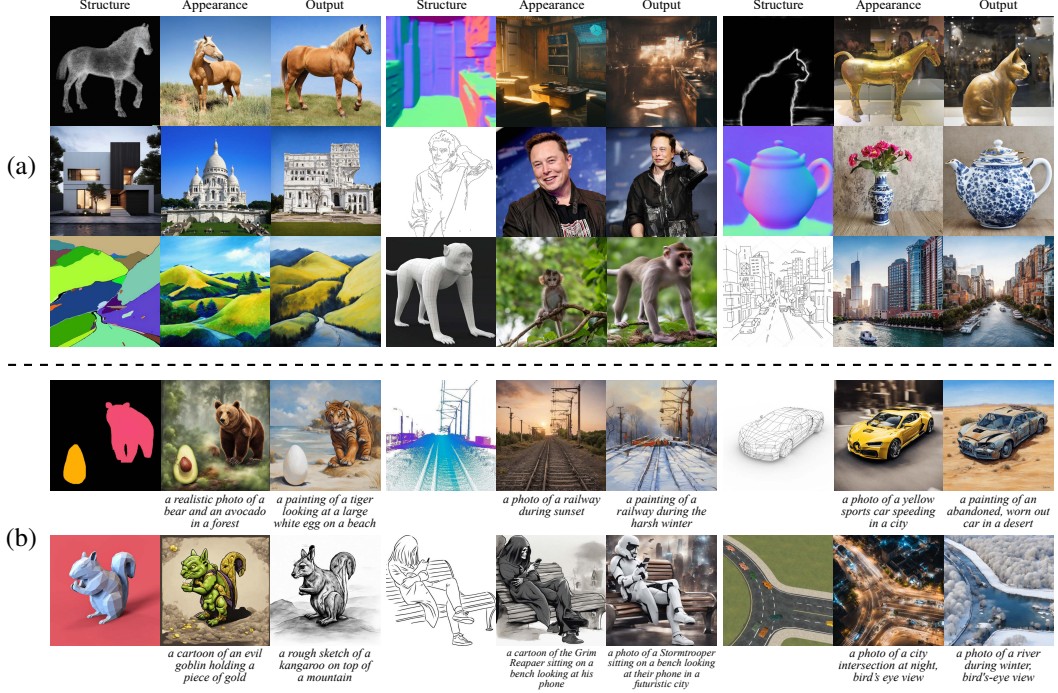

| Structure | Appearance | Output | Structure | Appearance | Output | Structure | Appearance | Output |

(a)

(b)

*a realistic photo of a bear and an avocado in a forest* — *a painting of a tiger looking at a large white egg on a beach* — *a photo of a railway during sunset* — *a painting of a railway during the harsh winter* — *a photo of a yellow sports car speeding in a city* — *a painting of an abandoned, worn out car in a desert*

*a cartoon of an evil goblin holding a piece of gold* — *a rough sketch of a kangaroo on top of a mountain* — *a cartoon of the Grim Reapaer sitting on a bench looking at his phone* — *a photo of a Stormtrooper sitting on a bench looking at their phone in a futuristic city* — *a photo of a city intersection at night, bird's eye view* — *a photo of a river during winter, bird's-eye view*

Figure 4: **Qualitative results for T2I diffusion structure and appearance control and conditional generation.** Ctrl-X supports a diverse variety of structure images for both (a) structure and appearance controllable generation and (b) prompt-driven conditional generation.

**Structure and appearance control.** Finally, we replace $\epsilon_\theta$ in Equation 2 with

$$\hat{\epsilon}_\theta \left( \mathbf{x}_t^\text{o} \mid t, \mathbf{c}, \{\mathbf{f}_{l,t}^\text{s}\}_{l \in L^\text{feat}}, \{\mathbf{A}_{l,t}^\text{s}\}_{l \in L^\text{self}}, \{\mathbf{h}_{l,t}^\text{a}\}_{l \in L^\text{app}} \right), \tag{8}$$

where $\{\mathbf{f}_{l,t}^\text{s}\}_{l \in L^\text{feat}}$, $\{\mathbf{A}_{l,t}^\text{s}\}_{l \in L^\text{self}}$, and $\{\mathbf{h}_{l,t}^\text{a}\}_{l \in L^\text{app}}$ respectively correspond to $\mathbf{x}_t^\text{s}$ features for feature injection, $\mathbf{x}_t^\text{s}$ attention maps for self-attention injection, and $\mathbf{x}_t^\text{a}$ features for appearance transfer.

## 5 Experiments

We present extensive quantitative and qualitative results to demonstrate the structure preservation and appearance alignment of Ctrl-X on T2I diffusion. Appendix A contains more implementation details.

### 5.1 T2I diffusion with structure and appearance control

**Baselines.** For training-based methods, ControlNet [44] and T2I-Adapter [25] learn an auxiliary module that injects a condition image into a pretrained diffusion model for structure alignment. We then combine them with IP-Adapter [43], a trained module for image prompting and thus appearance transfer. Uni-ControlNet [46] adds a feature extractor to ControlNet to achieve multi-image structure control of selected condition types, along with image prompting for global/appearance control. Splicing ViT Features [35] trains a U-Net from scratch per source-appearance image pair to minimize their DINO-ViT self-similarity distance and global [CLS] token loss. (For structure conditions not supported by a training-based baseline, we convert them to canny edge maps.) For guidance-based methods, FreeControl [24] enforce structure and appearance alignment via backpropagated score functions computed from diffusion feature subspaces. For guidance-free methods, Cross-Image Attention [1] manipulates attention weights to transfer appearance while maintaining structure. We run all methods on SDXL v1.0 [27] when possible and on their default base models otherwise.

**Dataset.** Our method supports T2I diffusion with appearance transfer and arbitrary-condition structure control. Since no benchmarks exist for such a flexible task, we create a new dataset comprising 256 diverse structure-appearance pairs. The structure images consist of 31% natural

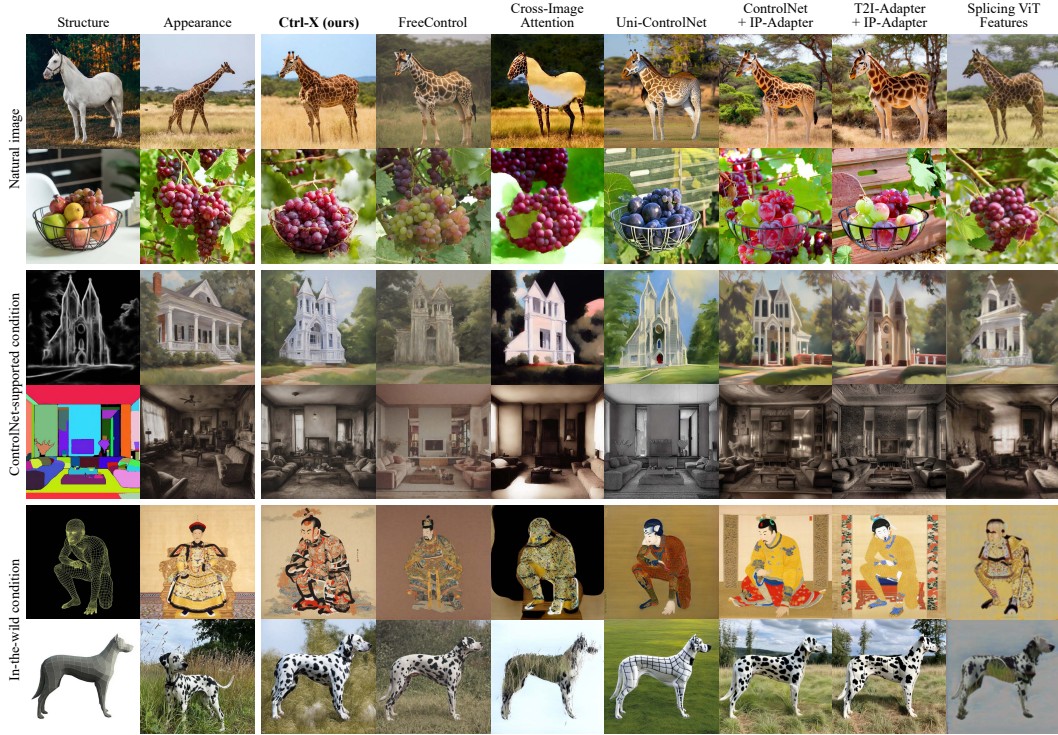

Figure 5: **Qualitative comparison of structure and appearance control.** Ctrl-X displays comparable structure control and superior appearance transfer compared to training-based methods. It is also more robust than guidance-based and guidance-free methods across diverse structure types.

images, $49\%$ ControlNet-supported conditions (*e.g.* canny, depth, segmentation), and $20\%$ in-the-wild conditions (*e.g.* 3D mesh, point cloud), and the appearance images are a mix of Web and generated images. We use templates and hand-annotation for the structure, appearance, and output text prompts.

**Evaluation metrics.** For quantitative evaluation, we report two widely-adopted metrics: *DINO Self-sim* measures the self-similarity distance [35] between the structure and output image in the DINO-ViT [6] feature space, where a lower distance indicates better structure preservation; *DINO-I* measures the cosine similarity between the DINO-ViT `[CLS]` tokens of the appearance and output images [30], where a higher score indicates better appearance transfer.

**Qualitative results.** As shown in Figures 4 and 5, Ctrl-X faithfully preserves structure from structure images ranging from natural images and ControlNet-supported conditions (*e.g.* HED, segmentation) to in-the-wild conditions (*e.g.* wireframe, 3D mesh) not possible in prior training-based methods while adeptly transferring appearance from the appearance image with semantic correspondence. Moreover, as shown in Figure 6, Ctrl-X is capable of multi-subject generation, capturing strong semantic correspondence between different subjects and the background, achieving balanced structure and appearance alignment. On the contrary, ControlNet + IP-Adapter [44, 43] often fails to maintain the structure and/or transfer the subjects' or background's appearances.

**Comparison to baselines.** Figure 5 and Table 2 compare Ctrl-X to the baselines for qualitative and quantitative results, respectively. Moreover, our user study in Table 4, Appendix A shows the human preference percentages of how often participants preferred Ctrl-X over each of the baselines on result quality, structure fidelity, appearance fidelity, and overall fidelity.

For training-based and guidance-based methods, despite Uni-ControlNet [46] and FreeControl's [24] stronger structure preservation (smaller DINO self-similarity), they generally struggle to enforce faithful appearance transfer and yield worse DINO-I scores, which is particularly visible in Figure 5 row 1 and 3. Since the training-based methods combine a structure control module (ControlNet [44] and T2I-Adapter [25]) with a separately-trained appearance transfer module IP-Adapter [43], the two modules sometimes exert conflicting control signals at the cost of appearance transfer (*e.g.* row 1)—and for ControlNet, structure preservation as well. For Uni-ControlNet, compressing the

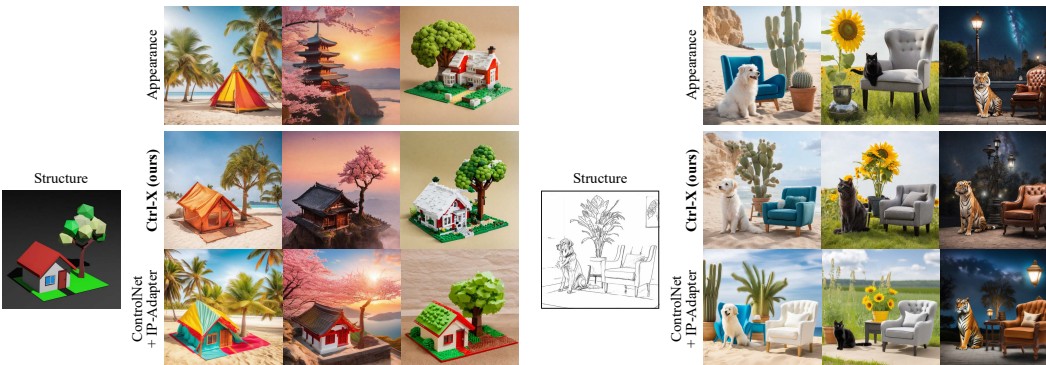

Figure 6: **Multi-subject generation.** Ctrl-X is capable of multi-subject generation with semantic correspondence between appearance and structure images across both subjects and backgrounds. In comparison, ControlNet + IP-Adapter [44, 43] often fails at transferring all subject appearances.

Table 1: **Inference efficiency.** Ctrl-X is slightly slower than training-based baselines yet significantly faster than training-free baselines and Splicing ViT Features. Moreover, Ctrl-X has lower peak GPU memory usage than SDXL v1.0 training-based methods and significantly lower memory than SDXL v1.0 training-free methods. (Uni-ControlNet and Cross-Image attention uses SD v1.5, which is $\sim 4$–$5\times$ faster and uses $\sim 3\times$ more memory compared to SDXL v1.0. Splicing ViT Features also trains its own much smaller custom model.)

| Method | Training | Preprocessing time (s) | Inference latency (s) | Total time (s) | Peak GPU memory usage (GiB) |
|---|---|---|---|---|---|
| Splicing ViT Features [35] | ✓ | 0.00 | 1557.09 | 1557.09 | 3.95 |
| Uni-ControlNet [46] | ✓ | 0.00 | 6.96 | 6.96 | 7.36 |
| ControlNet + IP Adapter [44, 43] | ✓ | 0.00 | 6.21 | 6.21 | 18.09 |
| T2I-Adapter + IP-Adapter [25, 43] | ✓ | 0.00 | 4.37 | 4.37 | 13.28 |
| Cross-Image Attention [1] | ✗ | 18.33 | 24.47 | 42.80 | 8.85 |
| FreeControl [24] | ✗ | 239.36 | 139.53 | 378.89 | 44.34 |
| **Ctrl-X (ours)** | ✗ | 0.00 | 10.91 | 10.91 | 11.51 |

appearance image to a few prompt tokens results in often inaccurate appearance transfer (*e.g.* rows 4 and 5) and structure bleed artifacts (*e.g.* row 6). For FreeControl, its appearance score function from extracted embeddings may not sufficiently capture more complex appearance correspondences, which, along with needing per-image hyperparameter tuning, results in lower contrast outputs and sometimes failed appearance transfer (*e.g.* row 4). Moreover, despite Splicing ViT Features [35] having the best self-similarity and DINO-I scores in Table 2, Figure 5 reveals that its output images are often blurry while displaying structure image appearance leakage with non-natural images (*e.g.* row 3, 5, and 6). It benchmarks well because its per-image training minimizes DINO metrics directly.

There is a trade-off between structure consistency (self-similarity) and appearance similarity (DINO-I), as these are competing metrics—increasing structure preservation corresponds to worse appearance similarity, which we show in Figure 11, Appendix B by varying controls schedules. As single metrics are not representative of overall method performance, we survey overall fidelity in our user study (Table 4, Appendix A), where Ctrl-X achieved best overall fidelity while matching result quality, structure fidelity, and appearance fidelity with training-based methods, showcasing our method's ability to balance the conflicting, disentangled tasks of structure and appearance control.

Guidance-free baseline Cross-Image Attention [1], in contrast, is less robust and more sensitive to the structure image, as the inverted structure latents contain strong appearance information. This causes both poorer structure alignment and frequent appearance leakage or artifacts (*e.g.* row 6) from the structure to the output images, resulting in worse DINO self-similarity and DINO-I scores. Similarly, Ctrl-X results are consistently preferred over Cross-Image Attention ones in our user study across all metrics (Table 4, Appendix A). In practice, we find Cross-Image Attention to be sensitive to its domain name, which is used for attention masking to isolate subjects, and it thus sometimes fails to produce outputs with cross-modal pairs (*e.g.* wireframes to photos).

**Inference efficiency.** We study the inference time, preprocessing time, and peak GPU memory usage of our method compared to the baselines, all with base model SDXL v1.0 except Uni-ControlNet (SD v1.5), Cross-Image Attention (SD v1.5), and Splicing ViT Features (U-Net). Table 1 reports the

Table 2: **Quantitative comparison of structure and appearance control.** Ctrl-X consistently outperforms both training-based and training-free methods in appearance alignment and shows comparable or better structure preservation compared to training-based and guidance-free methods, measured by DINO ViT self-similarity [35] and DINO-I [30], respectively.

| Method | Training | Natural image | | ControlNet-supported | | New condition | |
|---|---|---|---|---|---|---|---|
| | | Self-sim ↓ | DINO-I ↑ | Self-sim ↓ | DINO-I ↑ | Self-sim ↓ | DINO-I ↑ |
| Splicing ViT Features [35] | ✓ | 0.030 | 0.907 | 0.043 | 0.864 | 0.037 | 0.866 |
| Uni-ControlNet [46] | ✓ | **0.045** | 0.555 | **0.096** | 0.574 | **0.073** | 0.506 |
| ControlNet + IP-Adapter [44, 43] | ✓ | 0.068 | *0.656* | 0.136 | *0.686* | 0.139 | *0.667* |
| T2I-Adapter + IP-Adapter [25, 43] | ✓ | *0.055* | 0.603 | 0.118 | 0.586 | 0.109 | 0.566 |
| Cross-Image Attention [1] | ✗ | 0.145 | 0.651 | 0.196 | 0.510 | 0.175 | 0.570 |
| FreeControl [24] | ✗ | 0.058 | 0.572 | *0.101* | 0.585 | *0.089* | 0.567 |
| **Ctrl-X (ours)** | ✗ | 0.057 | **0.686** | 0.121 | **0.698** | 0.109 | **0.676** |

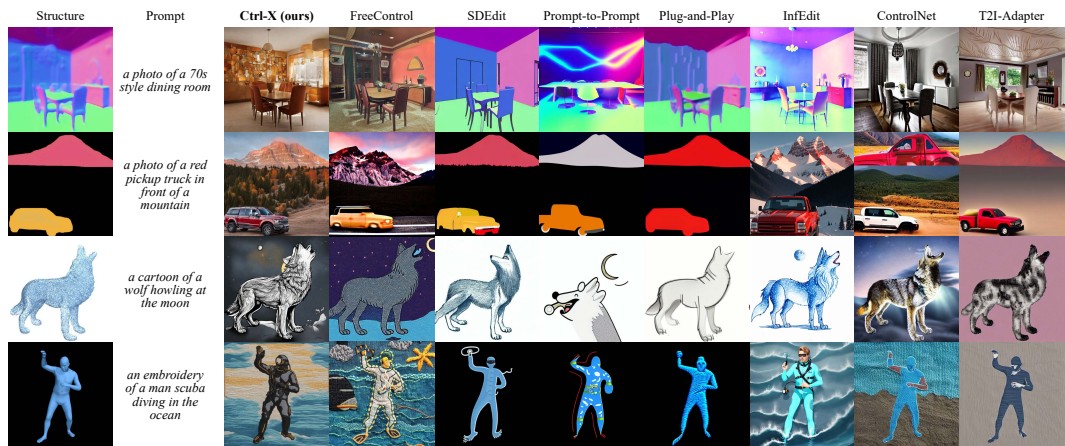

Figure 7: **Qualitative comparison of conditional generation.** Ctrl-X displays comparable structure control and superior prompt alignment to training-based methods, and it also has better image quality and is more robust than guidance-based and guidance-free methods across different conditions.

average inference time using a single NVIDIA H100 GPU. Ctrl-X is slightly slower than training-based ControlNet ($1.76\times$) and T2I-Adapter ($2.50\times$) with IP-Adapter yet significantly faster than per-image-trained Splicing ViT ($0.0070\times$), guidance-based FreeControl ($0.029\times$), and guidance-free Cross-Image Attention ($0.25\times$). Moreover, for methods with SDXL v1.0 as the base model, Ctrl-X has lower peak GPU memory usage than training-based methods and significantly lower memory than training-free methods. Our training-free and guidance-free method achieves comparable run time and peak GPU memory usage compared to training-based methods, indicating its flexibility.

**Extension to prompt-driven conditional generation.** Ctrl-X also supports prompt-driven conditional generation, where it generates an output image complying with the given text prompt while aligning with the structure from the structure image, as shown in Figures 4 and 7. Inspired by FreeControl [24], instead of a given $\mathbf{I}^a$, Ctrl-X can jointly generate $\mathbf{I}^a$ based on the text prompt alongside $\mathbf{I}^o$, where we obtain $\mathbf{x}_{t-1}^a$ via denoising with Equation 2 from $\mathbf{x}_t^a$ without control. Baselines, qualitative and quantitative analysis, and implementation details are available in Appendix C.

**Extension to video diffusion models.** Ctrl-X is training-free, guidance-free, and demonstrates competitive runtime. Thus, we can directly apply our method to text-to-video (T2V) models, as seen in Figure 17, Appendix D. Our method closely aligns the structure between the structure and output videos while transferring temporally consistent appearance from the appearance image.

## 5.2 Ablations

**Effect of control.** As seen in Figure 8(a), structure control is responsible for structure preservation (appearance-only *vs*. ours). Also, structure control alone cannot isolate structure information, display-

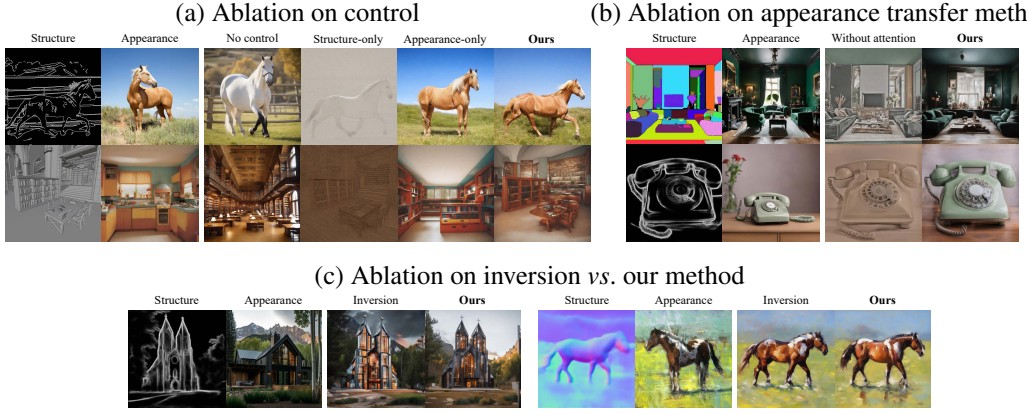

(a) Ablation on control  (b) Ablation on appearance transfer method

(c) Ablation on inversion *vs.* our method

Figure 8: **Ablations.**    We study ablations on control, appearance transfer method, and inversion.

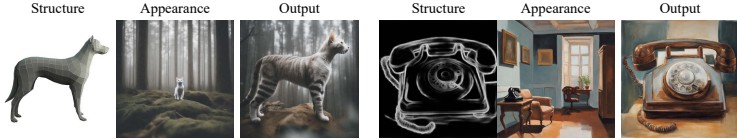

Figure 9: **Limitations.**    Ctrl-X can struggle with localizing the corresponding subject in the appearance image with appearance transfer when the subject is too small.

ing strong structure image appearance leakage and poor-quality outputs (structure-only *vs.* ours), as it merely injects structure features, which creates the semantic correspondence for appearance control.

**Appearance transfer method.**    As we consider appearance transfer as a stylization task, we compare our appearance statistics transfer with and without attention weighting in Figure 8(b). Without weighting (equivalent to AdaIN [15]), we have global normalization which ignores the semantic correspondence between the appearance and output images, so the outputs are low-contrast.

**Effect of inversion.**    We compare DDIM inversion *vs.* forward diffusion (ours) to obtain $\mathbf{x}_T^o = \mathbf{x}_T^s$ and $\mathbf{x}_t^s$ in Figure 8(c). Inversion displays appearance leakage from structure images in challenging conditions (left) while being similar to our method in others (right). Considering inversion costs and additional model inference time, forward diffusion is a better choice for our method.

## 6    Conclusion

We present Ctrl-X, a training-free and guidance-free framework for structure and appearance control of any T2I and T2V diffusion model. Ctrl-X utilizes pretrained T2I diffusion model feature correspondences, supports arbitrary structure image conditions, works with multiple model architectures, and achieves competitive structure preservation and superior appearance transfer compared to training- and guidance-based methods while enjoying the low overhead benefits of guidance-free methods. As shown in Figure 9, the key limitation of Ctrl-X is the semantic-aware appearance transfer method may fail to capture the target appearance when the instance is small because of the low resolution of the feature map. We hope our method and findings can unveil new possibilities and research on controllable generation as generative models become bigger and more capable.

**Broader impacts.**    Ctrl-X makes controllable generation more accessible and flexible by supporting multiple conditional signals (structure and appearance) and model architectures without the computational overhead of additional training or optimization. However, this accessibility also makes using pretrained T2I/T2V models for malicious applications (*e.g.* deepfakes) easier, especially since the controllability enables users to generate specific images and raises ethical concerns with consent and crediting artists for using their work as condition images. In response to these safety concerns, T2I and T2V models have become more secure. Likewise, Ctrl-X can inherit the same safeguards, and its plug-and-play nature allows the open-source community to scrutinize and improve its safety.

**Acknowledgements.**    This work was supported by the NSF Grants CCRI-2235012 and RI-2339769, the UCLA–Amazon Science Hub, and the Intel Rising Star Faculty Award.

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

# A    Method, implementation, and evaluation details

**More details on feed-forward structure control.**    We inject diffusion features *after* convolution skip connections. Since we initialize $\mathbf{x}_T^{\text{o}}$ as random Gaussian noise, the image structure after the first inference step likely does not align with $\mathbf{I}^{\text{s}}$, as observed by [36]. Thus, injecting *before* skip connections results in weaker structure control and image artifacts, as we are summing features $\mathbf{f}_t^{\text{o}}$ and $\mathbf{f}_t^{\text{s}}$ with conflicting structure information.

**More details on inference.**    With classifier-free guidance, inspired by [24, 1], we only control the prompt-conditioned $\epsilon_\theta$, 'steering' the diffusion process away from uncontrolled generation and thus strengthening structure and appearance alignment. Also, since structure and appearance control can result in out-of-distribution $\mathbf{x}_{t-1}$ after applying Equation 2, we apply $n^{\text{r}}$ steps of self-recurrence. Particularly, after obtaining $\mathbf{x}_{t-1}^{\text{o}}$ with structure and appearance control, we repeat

$$\mathbf{x}_{t-1}^{\text{o}} \leftarrow \sqrt{\alpha_{t-1}}\hat{\mathbf{x}}_0^{\text{o}} + \sqrt{1-\alpha_{t-1}}\hat{\epsilon}_\theta(\tilde{\mathbf{x}}_t^{\text{o}} \mid t, \mathbf{c}, \{\}, \{\}, \{\}),$$

$$\tilde{\mathbf{x}}_t^{\text{o}} := \sqrt{\frac{\alpha_t}{\alpha_{t-1}}}\mathbf{x}_{t-1}^{\text{o}} + \sqrt{1-\frac{\alpha_t}{\alpha_{t-1}}}\epsilon \quad \text{and} \quad \hat{\mathbf{x}}_0^{\text{o}} := \frac{\tilde{\mathbf{x}}_t^{\text{o}} - \sqrt{1-\alpha_t}\hat{\epsilon}_\theta(\tilde{\mathbf{x}}_t^{\text{o}} \mid t, \mathbf{c}, \{\}, \{\}, \{\})}{\sqrt{\alpha_t}} \quad (9)$$

$n^{\text{r}}$ times for (normalized) time steps $t \in [\tau_0^{\text{r}}, \tau_1^{\text{r}}]$, where $\tau_0^{\text{r}}, \tau_1^{\text{r}} \in [0,1]$. Notably, the self-recurrence steps occur *without* structure nor appearance control, and we observe generally lower artifacts and slightly better appearance transfer when self-recurrence is enabled.

**Comparison to prior works.**    We compare Ctrl-X to prior works in terms of capabilities in Table 3. Compared to baselines, our method is the only work which supports appearance and structure control with any structure conditions, while being training-free and guidance-free.

**Experiment hyperparameters.**    For both T2I diffusion with structure and appearance control and structure-only conditional generation, we use Stable Diffusion XL (SDXL) v1.0 [27] for all Ctrl-X experiments, unless stated otherwise. For SDXL, we set $L^{\text{feat}} = \{0\}_{\text{decoder}}$, $L^{\text{self}} = \{0, 1, 2\}_{\text{decoder}}$, $L^{\text{app}} = \{1, 2, 3, 4\}_{\text{decoder}} \cup \{2, 3, 4, 5\}_{\text{encoder}}$, and $\tau^{\text{s}} = \tau^{\text{a}} = 0.6$. We sample $\mathbf{I}^{\text{o}}$ with 50 steps of DDIM sampling and set $\eta = 1$ [33], doing self-recurrence for $n^{\text{r}} = 2$ for $\tau_0^{\text{r}} = 0.1$ and $\tau_1^{\text{r}} = 0.5$. We implement Ctrl-X with `Diffusers` [37] and run all experiments on a single NVIDIA A6000 GPU, except evaluating inference efficiency in Table 1 where we run on a single NVIDIA H100 GPU.

**More details on evaluation metrics.**    To evaluate structure and appearance control results (Table 2), we report DINO Self-sim and DINO-I. For DINO Self-sim, we compute the self-similarity (i.e., mean squared error) between the structure and output image in the DINO-ViT [6] feature space, where we use the base-sized model with patch size 8 following Splicing ViT Features [35]. For DINO-I, we compute the cosine similarity between the DINO-ViT `[CLS]` tokens of the appearance and output images, where we use the small-sized model with patch size 16 following DreamBooth [30].

To evaluate prompt-driven controllable generation results (Table 5), we report DINO-Self-sim, CLIP score, and LPIPS. DINO Self-sim is computed the same way as structure and appearance control metrics. For CLIP score, we compute the cosine similarity between the output image and text prompt in the CLIP embedding space, where we use the large-sized model with patch size 14 (ViT-L/14) following FreeControl [24]. For LPIPS, we compute the appearance deviation of the output image from the structure image, where we use the official `lpips` package [45] with AlexNet (`net="alex"`).

**User study.**    We follow the setting of the user study from DenseDiffusion [17], where we compare Ctrl-X to baselines on structure and appearance control in Table 4, we display the average human preference percentages of how often participants preferred our method over each of the baselines. We randomly selected 15 sample pairs from our dataset and then assigned each sample pair to 7 methods: Splicing ViT Feature [35], Uni-ControlNet [46], ControlNet + IP-Adapter [44, 43], T2I-Adapter + IP-Adapter [25, 43], Cross-Image Attention [1], FreeControl [24], and Ctrl-X. We invited 10 users to evaluate pairs of results, each consisting of our method, Ctrl-X, and a baseline method. For each comparison, users assessed 15 pairs between Ctrl-X and each baseline, based on four criteria: "the quality of displayed images," "the fidelity to the structure reference," "the fidelity to the appearance reference," and "overall fidelity to both structure and appearance reference," which we denote result quality, structure fidelity, appearance fidelity, and overall fidelity, respectively. We collected 150 comparison results for between Ctrl-X and each individual baseline method. We reported the human preference rate, which indicates the percentage of times participants preferred our results over the baselines. The user study demonstrates that Ctrl-X outperforms training-free baselines and has a competitive performance compared to training-based baselines.

Table 3: **Comparison to prior works.** Comparing the capabilities of Ctrl-X to prior controllable generation works. Natural images and in-the-wild conditions refer to the type of structure image that the method supports for structure control.

| Method | Structure control | | Appearance control | Training-free | Guidance-free |
|---|---|---|---|---|---|
| | *Natural images* | *In-the-wild conditions* | | | |
| Uni-ControlNet [46] | | | ✓ | | ✓ |
| ControlNet [44] (+ IP-Adapter [43]) | | | ✓ | | ✓ |
| T2I-Adapter [25] (+ IP-Adapter [43]) | | | ✓ | | ✓ |
| SDEdit [23] | ✓ | | | ✓ | ✓ |
| Prompt2Prompt [11] | ✓ | | | ✓ | ✓ |
| Plug-and-Play [36] | ✓ | | | ✓ | ✓ |
| InfEdit [41] | ✓ | | | ✓ | ✓ |
| Splicing ViT Attention [35] | ✓ | | ✓ | | ✓ |
| Cross-Image Attention [1] | ✓ | | ✓ | | ✓ |
| FreeControl [24] | ✓ | ✓ | ✓ | ✓ | |
| **Ctrl-X (ours)** | ✓ | ✓ | ✓ | ✓ | ✓ |

Table 4: **Qualitative comparison of structure and appearance control via user study.** The human preference percentages here show how often the participants preferred Ctrl-X over each of the baselines on result quality, structure fidelity, appearance fidelity, and overall fidelity. Ctrl-X consistently outperforms training-free baselines and is competitive with training-based ones, especially with overall fidelity, showcasing Ctrl-X's ability to balance structure and appearance control.

| Method | Training | Result quality ↑ | Structure fidelity ↑ | Appearance fidelity ↑ | Overall fidelity ↑ |
|---|---|---|---|---|---|
| Splicing ViT Features [35] | ✓ | 95% | 87% | 56% | 78% |
| Uni-ControlNet [46] | ✓ | 86% | 17% | 96% | 74% |
| ControlNet + IP-Adapter [44, 43] | ✓ | 46% | 61% | 41% | 50% |
| T2I-Adapter + IP-Adapter [25, 43] | ✓ | 74% | 53% | 67% | 58% |
| Cross-Image Attention [1] | ✗ | 95% | 83% | 83% | 83% |
| FreeControl [24] | ✗ | 64% | 48% | 79% | 74% |
| **Ctrl-X (ours)** | ✗ | - | - | - | - |

The user study (Figure 10) is conducted via Amazon Mechanical Turk.

# B  Structure and appearance schedules and higher-level conditions

Ctrl-X has two hyperparameters, structure control schedule ($\tau^\text{s}$) and appearance control schedule ($\tau^\text{a}$), which enable finer control over the influence of the structure and appearance images on the

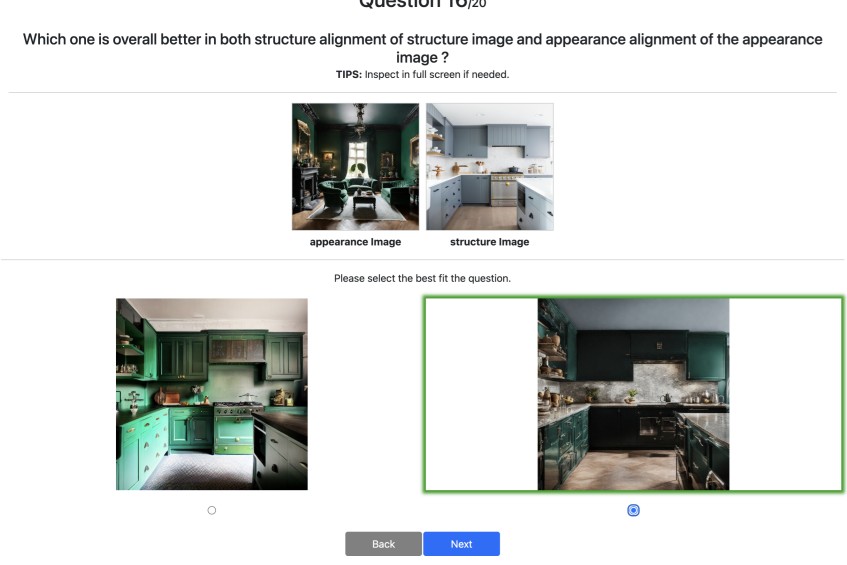

Figure 10: **User study interface.** A screenshot of our user study's interface.

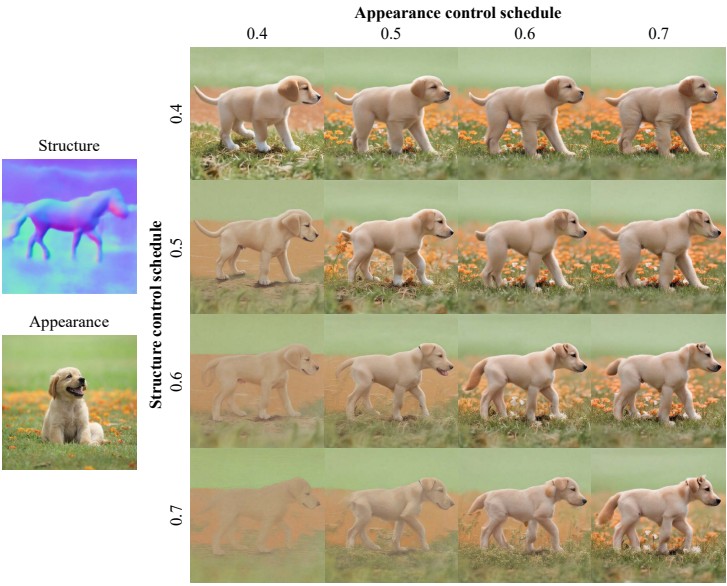

Figure 11: **Ablation on control schedules.** By varying Ctrl-X's structure and appearance control schedules ($\tau^\mathrm{s}$ and $\tau^\mathrm{a}$), we change the influence of the structure and appearance images on the output.

output. As structure alignment and appearance transfer are conflicting tasks, controlling the two schedules allows the user to determine the best tradeoff between the two. The default values of $\tau^\mathrm{s} = 0.6$ and $\tau^\mathrm{a} = 0.6$ we choose merely works well for most—but not all—structure-appearance image pairs. Particularly, this control enables better results for challenging structure-appearance pairs and allows our method to be used with higher-level conditions without clear subject outlines.

**Effect of control schedules.** We vary structure and appearance control schedules ($\tau^\mathrm{s}$ and $\tau^\mathrm{a}$) as seen in Figure 11. Decreasing structure control can make cross-class structure-appearance pairs (e.g., horse normal map with puppy appearance) look more realistic, as doing so trades strict structure adherence for more sensible subject shapes in challenging scenarios. Decreasing appearance control trades appearance alignment for less artifacts. Note that, generally, $\tau^\mathrm{s} \leq \tau^\mathrm{a}$, as structure control requires appearance transfer to realize the structure information and avoid structure image appearance leakage, most prominently demonstrated in Figure 8(a).

**Higher-level structure conditions.** By decreasing the structure control schedule $\tau^\mathrm{s}$ from the default 0.6 to 0.3–0.5, Ctrl-X can handle sparser and higher-level structure conditions such as bounding boxes and human post skeletons/keypoints, shown in Figure 12. Not only does this make our method applicable to other higher-level control types, it also generally reduces structure image appearance leakage with challenging structure conditions.

## C   Extension to prompt-driven controllable generation

Ctrl-X also supports prompt-driven conditional generation, where it generates an output image complying with the given text prompt while aligning with the structure from the structure image, as shown in Figures 4 and 7. Inspired by FreeControl [24], instead of a given $\mathbf{I}^\mathrm{a}$, Ctrl-X can jointly generate $\mathbf{I}^\mathrm{a}$ based on the text prompt alongside $\mathbf{I}^\mathrm{o}$, where we obtain $\mathbf{x}_{t-1}^\mathrm{a}$ via denoising with Equation 2 from $\mathbf{x}_t^\mathrm{a}$ without control.

**Baselines.** For training-based methods, we test ControlNet [44] and T2I-Adapter [25]. For guidance-based methods, we test FreeControl [24], where we generate an appearance image alongside the output image instead of inverting a given appearance image. For guidance-free methods, SDEdit [23] adds noise to the input image and denoises it with a pretrained diffusion model to preserve structure. Prompt-to-Prompt [11] and Plug-and-Play [36] manipulate features and attention of pretrained T2I models for prompt-driven image editing. InfEdit [41] uses three-branch attention manipulation and consistent multi-step sampling for fast, consistent image editing.

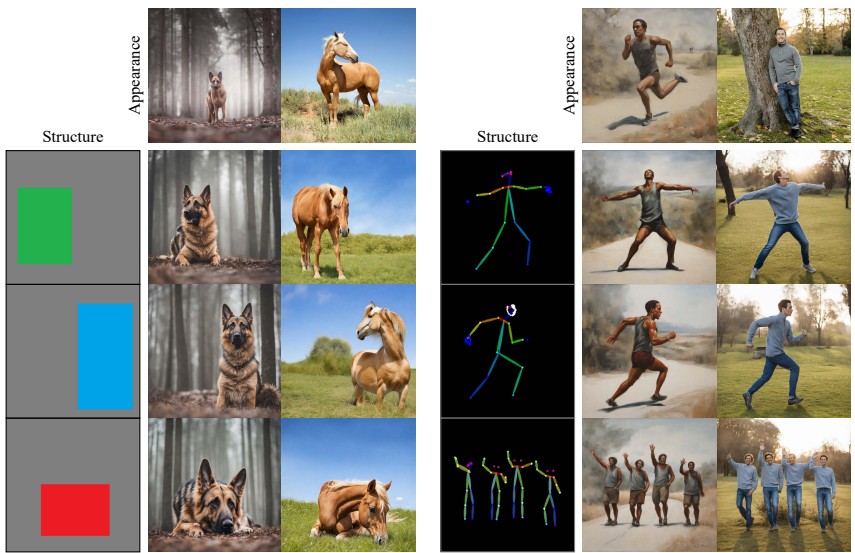

Figure 12: **Higher-level structure conditions.** By decreasing the structure schedule $\tau^s$ (from the default 0.6 to 0.3–0.5), Ctrl-X can handle higher-level structure conditions such as bounding boxes (left) and human pose skeletons/keypoints (right).

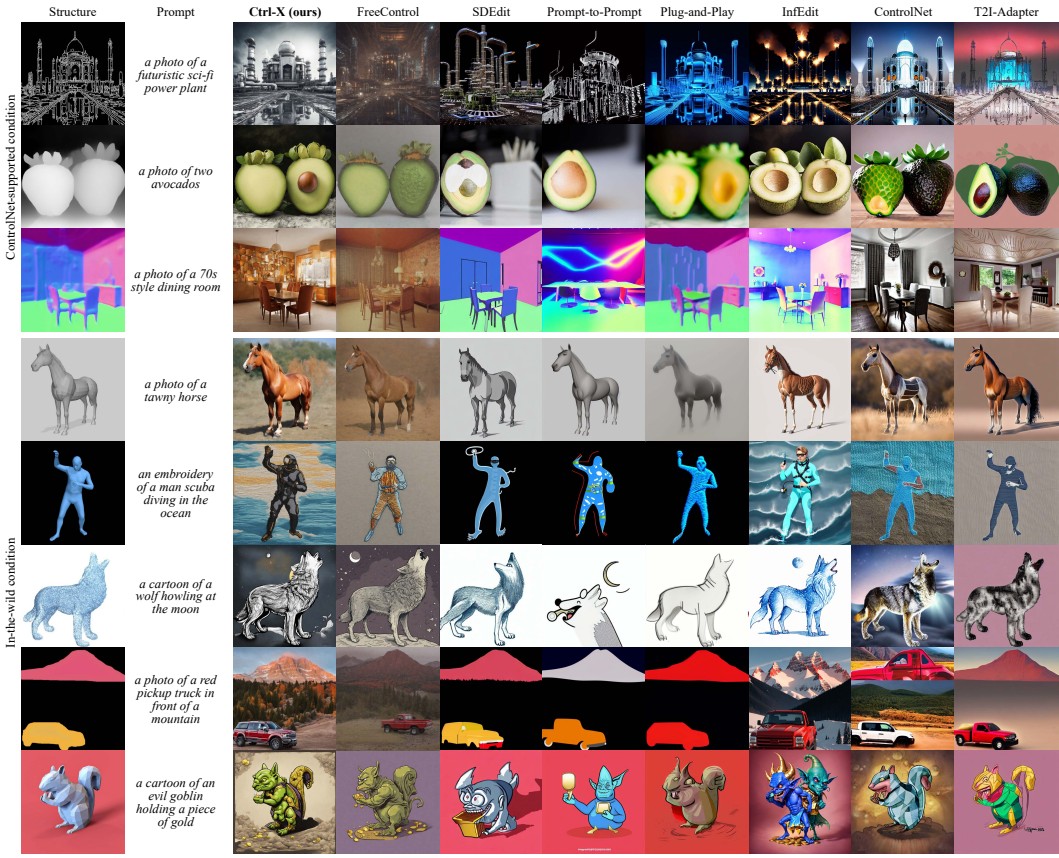

Figure 13: **Full qualitative comparison of conditional generation.** Ctrl-X displays comparable structure control and superior prompt alignment to training-based methods with better image quality. It is also more robust than guidance-based and guidance-free methods across a wide variety of condition types. (We run ControlNet [44] and T2I-Adapter [25] on SD v1.5 [29] instead of SDXL v1.0 [27], as the latter frequently generates low-contrast, flat results for the two methods.)

Table 5: **Quantitative comparison on conditional generation.** Ctrl-X outperforms all training-based and guidance-free baselines in prompt alignment (CLIP score). Although many baselines seem to better preserve structure with low DINO self-similarity distances, the low distances mainly come from severe structure image appearance leakage (high LPIPS), also shown in Figure 13. Also, though FreeControl displays better structure preservation and prompt alignment, it still experiences appearance leakage which results in poor image quality (Figure 13).

| Method | Training | ControlNet-supported | | | New condition | | |
|---|---|---|---|---|---|---|---|
| | | Self-sim ↓ | CLIP score ↑ | LPIPS ↑ | Self-sim ↓ | CLIP score ↑ | LPIPS ↑ |
| ControlNet [44] | ✓ | 0.126 | 0.298 | 0.657 | 0.092 | 0.302 | 0.507 |
| T2I-Adapter [25] | ✓ | 0.096 | 0.303 | 0.504 | 0.068 | 0.302 | 0.415 |
| SDEdit [23] | ✗ | 0.102 | 0.300 | 0.366 | 0.096 | 0.309 | 0.373 |
| Prompt-to-Prompt [11] | ✗ | 0.100 | 0.276 | 0.370 | 0.097 | 0.287 | 0.357 |
| Plug-and-Play [36] | ✗ | 0.056 | 0.282 | 0.272 | 0.050 | 0.292 | 0.301 |
| InfEdit [41] | ✗ | 0.117 | 0.314 | 0.523 | 0.102 | 0.311 | 0.442 |
| FreeControl [24] | ✗ | 0.108 | 0.340 | 0.557 | 0.104 | 0.339 | 0.492 |
| **Ctrl-X (ours)** | ✗ | 0.134 | 0.322 | 0.635 | 0.135 | 0.326 | 0.590 |

**Dataset.** Our controllable generation dataset comprises of 175 diverse image-prompt pairs with the same (structure) images as Section 5.1. It consists of 71% ControlNet-supported conditions and 29% new conditions. We use the same hand-annotated structure prompts and hand-create output prompts with inspiration from Plug-and-Play's datasets [36]. See more details in Appendix E.

**Evaluation metrics.** For quantitative evaluation, we report three widely-adopted metrics: *DINO Self-sim* from Section 5.1 measures structure preservation; *CLIP score* [28] measures the similarity between the output image and text prompt in the CLIP embedding space, where a higher score suggests stronger image-text alignment; *LPIPS* distance [45] measures the appearance deviation of the output image from the structure image, where a higher distance suggests lower appearance leakage from the structure image.

**Qualitative results.** As shown in Figures 4 and 13, Ctrl-X generates high-quality images with great structure preservation and close prompt alignment. Our method can extract structure information from a wide range condition types and produces results of diverse modalities based on the prompt.

**Comparison to baselines.** Figure 7 and Table 5 compare our method to the baselines. Training-based methods typically better preserve structure, with lower DINO self-similarity distances, at the cost of worse prompt adherence, with lower CLIP scores. This is because these modules are trained on condition-output pairs which limit the output distribution of the base T2I model, especially for in-the-wild conditions where the produced canny maps are unusual. Our method, in contrast, transfers appearance from a jointly-generated appearance image that utilizes the full generation power of the base T2I model and is neither domain-limited by training nor greatly affected by hyperparameters.

In contrast, guidance-based and guidance-free methods display appearance leakage from the structure image. The guidance-based FreeControl requires per-image hyperparameter tuning, resulting in fluctuating image quality and appearance leakage when ran with its default hyperparameters. Thus, even if it displays slightly higher prompt adherence (higher CLIP score), the appearance leakage often produces lower-quality output images (lower LPIPS). Guidance-free methods, on the other hand, share (inverted) latents (SDEdit, Prompt-to-Prompt, Plug-and-Play) or injects diffusion features (all) with the structure image without the appearance regularization which Ctrl-X's jointly-generated appearance image provides. Consequently, though structure is preserved well with better DINO self-similarity distances, undesirable structure image appearance is also transferred over, resulting in worse LPIPS scores. For example, all guidance-based and guidance-free baselines display the magenta-blue-green colors of the dining room normal map (row 3), the color-patchy look of the car and mountain sparse map (row 7), and the red background of the 3D squirrel mesh (row 8).

# D  Additional results

**Additional structure and appearance control results.** We present additional results of structure and appearance control in Figure 14.

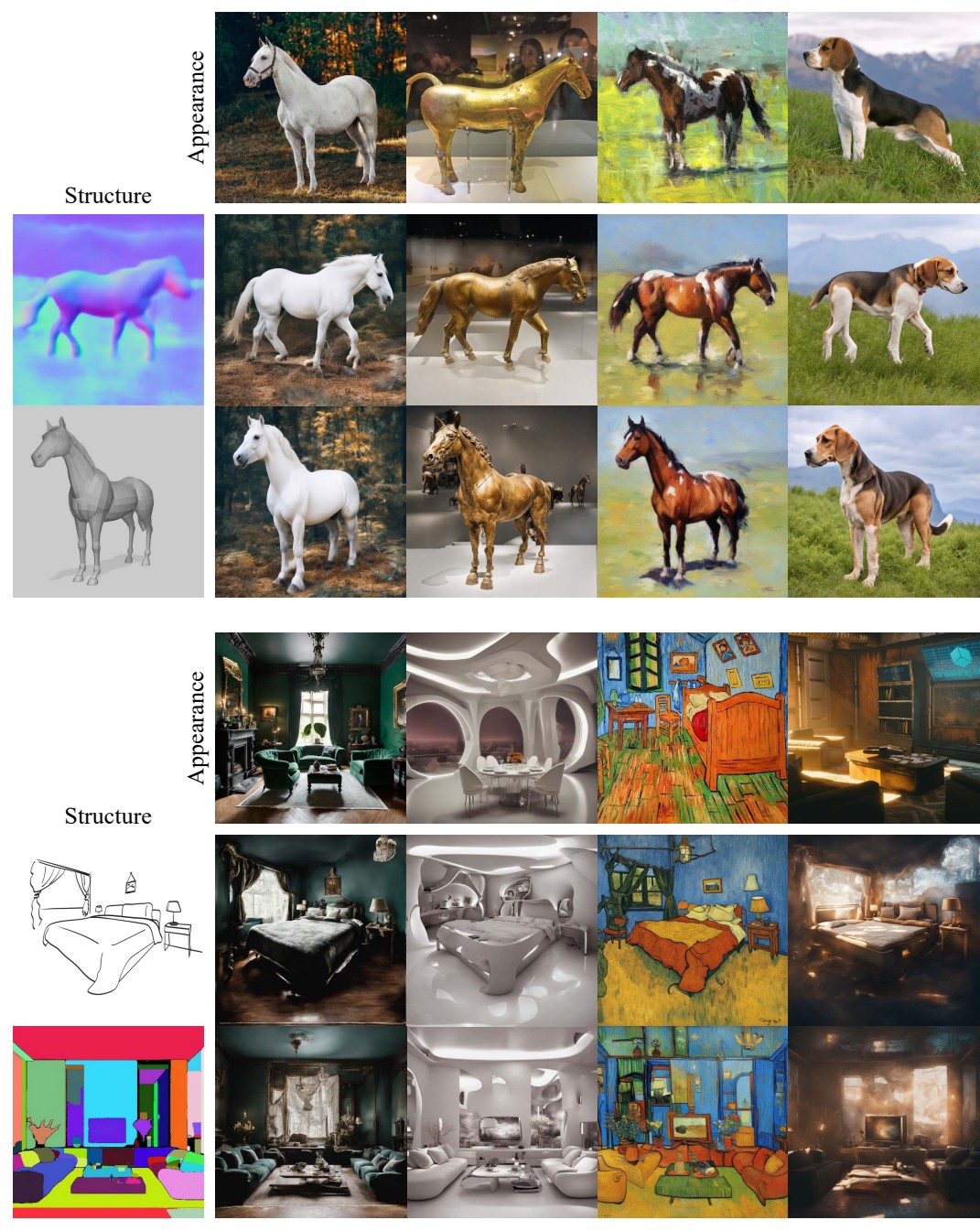

Figure 14: **Additional results of structure and appearance control.** We present additional Ctrl-X results of structure and appearance control.

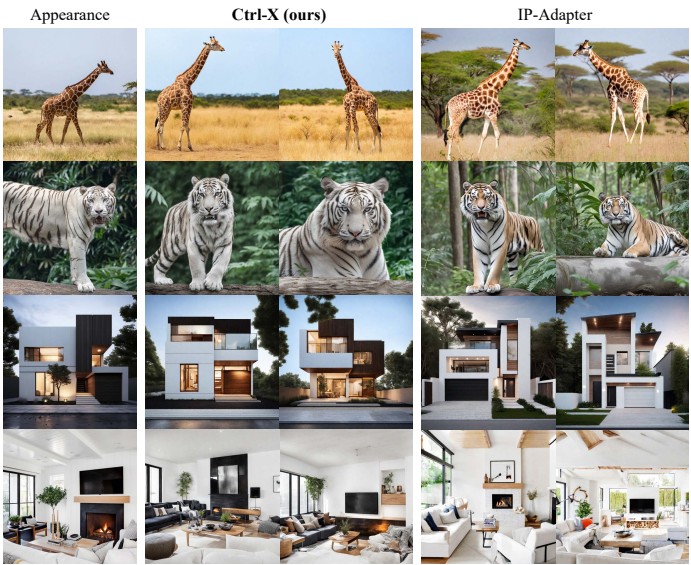

Figure 15: **Appearance-only control.** Ctrl-X can do appearance-only control by dropping the structure control branch. Compared to IP-Adapter [43], our method shows better appearance alignment for both subjects and backgrounds.

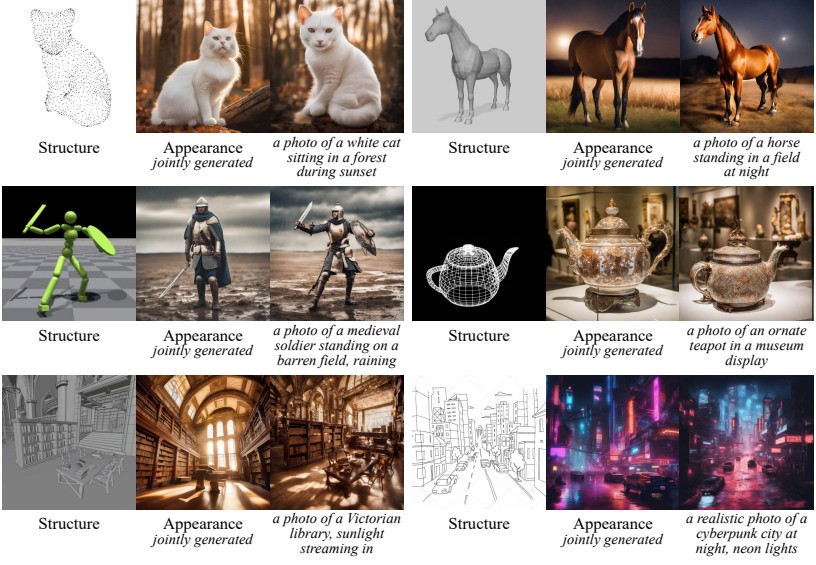

Figure 16: **Structure-only control.** We display the jointly generated appearance images for prompt-driven conditional generation. Ctrl-X appearance transfer preserves the image quality of the generated appearances, so structure-only retains the quality of the base model.

**Appearance-only control.** Ctrl-X is a method which disentangles control from given structure and appearance images, balancing structure alignment and appearance transfer when the two tasks are inherently conflicting. However, Ctrl-X can also achieve appearance-only control by simply dropping the structure control branch (and thus not needing to generate a structure image), as shown in Figure 15. Our method displays better appearance alignment for both subjects and background compared to the training-based IP-Adapter [43].

**Structure-only control.** For prompt-driven conditional (structure-only) generation, Ctrl-X needs to jointly generate an appearance image, where the jointly generated image is equivalent to vanilla SDXL v1.0 generation. We display the outputs alongside these appearance images in Figure 16, where there is minimal quality difference between the generated appearance images and the appearance-

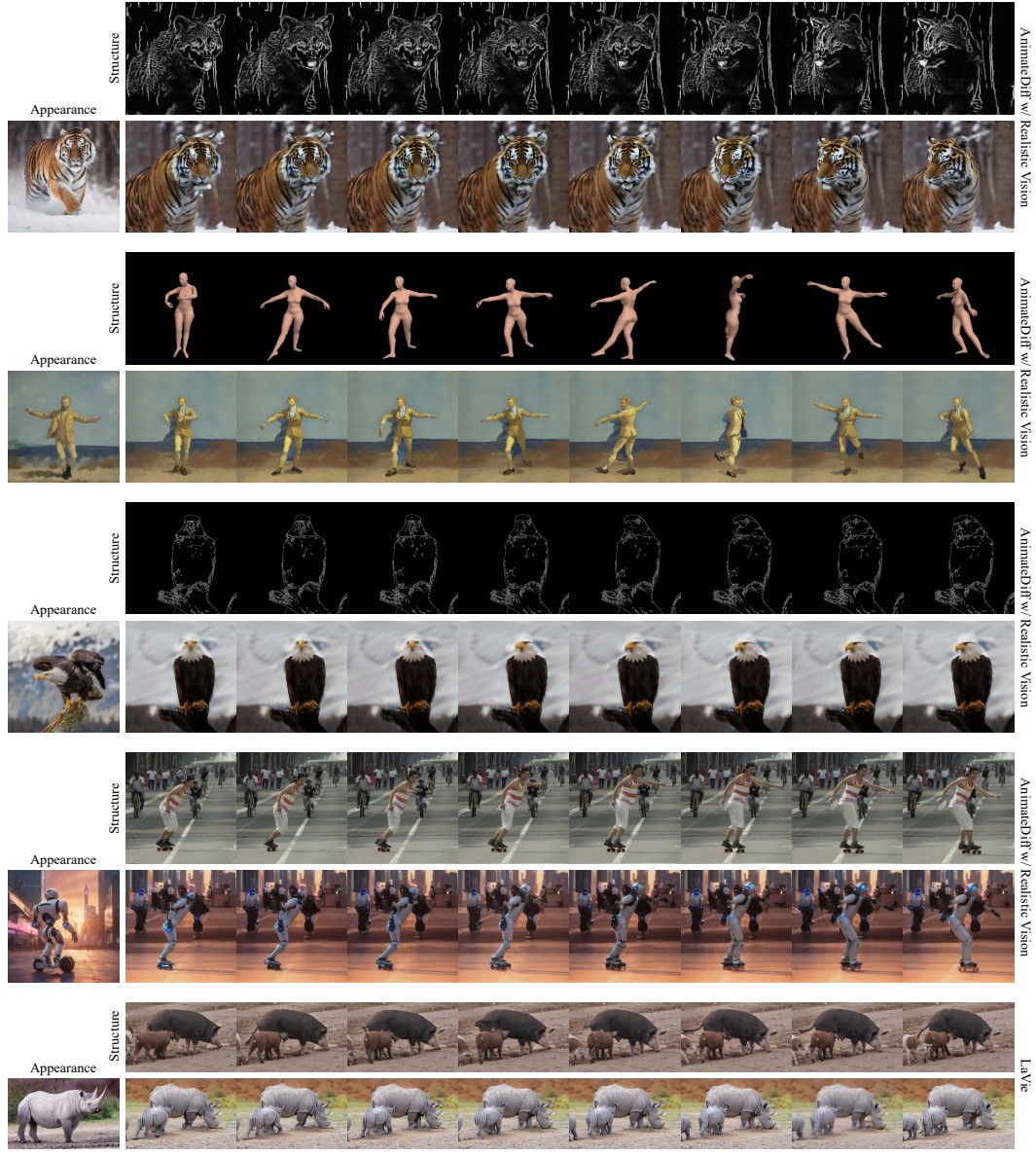

Figure 17: **Extension to text-to-video (T2V) models.** Ctrl-X can be directly applied to T2V models for controllable video structure and appearance control, with AnimateDiff [9] with Realistic Vision v5.1 [32] and LaVie [39] here as examples. A playable video version of the AnimateDiff results can be found in the attached supplementary `zip` file as `ctrl_x_animatediff.mp4`.

transferred output images, indicating that the need for appearance transfer does not greatly impact image quality. Thus, Ctrl-X adheres well to the quality of its base models.

**Extension to video diffusion models.** We also present results of our method directly applied to text-to-video (T2V) diffusion models in Figure 17, namely AnimateDiff [9] with base model Realistic Vision v5.1 [32] and LaVie [39]. A playable video version of the AnimateDiff T2V results can be found in the attached supplementary `zip` file as `ctrl_x_animatediff.mp4`.

# E  Dataset details

For our dataset, we list all images present in the paper and their associated sources and licenses present in the paper in `dataset_sources.pdf` in the supplementary materials `zip`. All academic

datasets which we use are cited here [3, 10, 43, 24, 48, 22, 19, 36, 16]. We publicly release our dataset in our code release: `https://github.com/genforce/ctrl-x`.

**Overview.**    Our dataset consists of 177 $1024 \times 1024$ images divided into 16 types and across 7 categories. We split the images into condition images (67 images: "canny edge map", "metadrive", "3d mesh", "3d humanoid", "depth map", "human pose image", "point cloud", "sketch", "line drawing", "HED edge drawing", "normal map", and "segmentation mask") and natural images (110 images: "photo", "painting", "cartoon" and "birds eye view"), with the the largest type being "photo" (83 images). The condition images are further divided into two groups in our paper: ControlNet-supported conditions ("canny edge map", "depth map", "human pose image", "line drawing", "HED edge drawing", "normal map", and "segmentation mask") and in-the-wild conditions ("metadrive", "3D mesh", "3D humanoid", "point cloud", and "sketch"). All of our images fall into one of seven categories: "animals" (52 images), "buildings" (11 images), "humans" (28 images), "objects" (29 images), "rooms" (24 images), "scenes" (22 images) and "vehicles" (11 images). About two thirds of the images come from the Web, while the remaining third is generated using SDXL 1.0 [27] or converted from natural images using Controlnet Annotators packaged in `controlnet-aux` [44]. For each of these images, we hand annotate them with a text prompt and other metadata (*e.g.* type). Then, these images, promtps, and metadata are combined to form the structure and appearance control dataset and conditional generation dataset, detailed below.

**T2I diffusion with structure and appearance control dataset.**    This dataset consists of 256 pairs of images from the image dataset described above. This dataset is used to evaluate our method and the baselines' ability to generate images adhering to the structure of a condition or natural image while aligning to the appearance of a second natural image. Each pair contains a structure image (which may be a condition or natural image) and an appearance image (which is a natural image). The dataset also includes a structure prompt for the structure image (*e.g.* "a canny edge map of a horse galloping"), an appearance prompt for the appearance image (*e.g.* "a painting of a tawny horse in a field"), and one target prompt for the output image (*e.g.* "a painting of tawny horse galloping") generated by combining the metadata of the appearance and structure prompts via a template, with a few edge cases hand-annotated. Image pairs are constructed from two images from the same category (*e.g.* "animals") and the majority of pairs consist of images of the same subject (*e.g.* "horse"), but we include 30 pairs of cross-subject images (*e.g.* "cat" and "dog") to test the methods' ability to generalize structure information across subjects.

In practice, when running Ctrl-X, we set the appearance prompt to be the same as the output prompt instead of our hand-annotated appearance prompt. We found little differences between the two.

**Conditional generation dataset.**    The conditional dataset combines conditional images with both template-generated and hand-written output prompts (inspired by Plug-and-Play [36] and FreeControl [24]) to evaluate our method and the baselines' ability to construct an image adhering to the structure of the input image while complying with the given prompt. Each entry in the conditional dataset consists of a condition image combined with a unique prompt. We have 175 such condition-prompt pairs from the set of 66 condition images above.

