# OpenReview forum: "Ctrl-X: Controlling Structure and Appearance for Text-To-Image Generation Without Guidance"
_NeurIPS.cc/2024/Conference — NeurIPS 2024 poster_

### Official Review · Reviewer_YZfZ · 2024-07-12

**Soundness:** 2
**Presentation:** 3
**Contribution:** 2
**Rating:** 5
**Confidence:** 3

**Summary:**

This paper presents a training-free and guidance-free method for controllable image/video generation with structure and appearance control. Specifically, Ctrl-X injects structural and appearance features directly into the noised samples via cross-attention. Compared with other baselines for structural and appearance control, the proposed method achieves good appearance alignment and structure presentation.

**Strengths:**

The proposed method is a good complement to training-based and guidance-based controllable visual generation methods. Experiment results also shown the effectiveness of the proposed method.

**Weaknesses:**

- It looks like the generated images/videos are a little bit painting style. I guess this is caused by injecting structure features directly without training/fine-tuning.

- The training-based baselines are not very strong. Combining two existing models (e.g., ControlNet+IP-Adapter, T2I-Adapter+IP-Adapter) might not be a fair comparison, since they are not specifically designed for both appearance and structure control. I would suggest compare structural preservation ability with ControlNet (SD1/2/3, SDXL) and T2I-Adapter, and then compare appearance control ability with IP-Adapter.

**Questions:**

My understanding is that for training-free methods, we are making a trade-off between control quality and training computation cost. There exists some efficient training-based methods for controllable generation [1,2]. A natural question is: compared with such training-based method, is the training-free property good enough for us to use Ctrl-X?


[1] Ran, Lingmin, et al. "X-adapter: Adding universal compatibility of plugins for upgraded diffusion model." Proceedings of the IEEE/CVF Conference on Computer Vision and Pattern Recognition. 2024.

[2] Lin, Han, et al. "Ctrl-Adapter: An Efficient and Versatile Framework for Adapting Diverse Controls to Any Diffusion Model." arXiv preprint arXiv:2404.09967 (2024).

---

> ### Author Rebuttal · Authors · 2024-08-06
>
> Thank you for your time and feedback on our work. We address your suggestions and concerns below. **You can find the referred figures in the PDF attached with the “global” response. Tables can similarly be found in the main body of the “global” response.**
>
> **Painting style generation results.**
> Our proposed method works for any domain. Many realistic and non-painting samples generated are included in Figure 1, 4, 5 and Supplementary Figures 10, 11 *in the paper*. We will update the paper to make everything clear. Moreover, many samples in the attached rebuttal PDF also have realistic generation results—if the base model (e.g., SDXL) can generate realistic results for a prompt (which SDXL almost always can), then our method can, too.
>
> **Comparing with training-based baselines for individual structure or appearance control.**
> Thanks for the suggestion!  For appearance-only control, we compare Ctrl-X (by dropping structure control) against IP-Adapter in Figure 4, where Ctrl-X displays better appearance alignment than IP-Adapter for both the subjects and backgrounds. For structure-only control, we compare Ctrl-X against both training-based baselines (ControlNet and T2I-Adapter) and training-free baselines (FreeControl, Plug-and-Play, etc.) in our Appendix, specifically Figure 10 and Table 3 *in the paper Appendix*.
>
> Additionally, we present additional experimental results with Uni-ControlNet [1], a ControlNet-based model trained to control structure and appearance at the same time. We report the user study and quantitative results in Tables 1 and 2. Ctrl-X is significantly ahead of Uni-ControlNet in terms of human preference for overall fidelity, indicating our method’s ability to balance structure and appearance control. Moreover, though Uni-ControlNet has better DINO self-sim, it struggles to balance structure preservation with appearance transfer with worse DINO-I scores, which is echoed by the user study.
>
> **Trade-off between training-time and quality.**
> Thanks for the thought-provoking discussion starter. Ctrl-X provides a training-free and optimization-free method that achieves a delicate balance between structure control and appearance transfer. Though the argument that adapters have made it less necessary to train control modules from scratch is totally valid, the training cost (or backpropagation cost for guidance-based methods) cannot be ignored as we need to backpropagate through new models that will only get bigger. For example, Ctrl-Adapter [2] is trained on 200K images—paired data for any condition that Ctrl-X supports is difficult to gather at this scale. Also, FreeControl (being guidance-based) requires ~44GB of VRAM on SDXL to run, compared to Ctrl-X’s 11.5GB VRAM usage (Table 3).
>
> The flexibility of training-free methods is also important, as Ctrl-X works with a large range of structure control signals (including higher-level conditions like bounding box and pose skeleton as shown in Figure 3), which training-based methods are limited by, as training requires paired data difficult to gather for in-the-wild conditions like 3D mesh, point cloud, etc. Thus, Ctrl-X’s training-free property has many upsides that may not necessarily make it a “trade-off.”
>
> Of course, training-based methods still excel at the specific tasks they are trained for—a canny ControlNet, for example, is great at canny-image-conditioned generation. However, their limited flexibility makes the “trade-off” of Ctrl-X’s training-free nature a lot more appealing, as Ctrl-X works for a much wider range of applications, condition types or models. This may be closer to what we ultimately want for “controllable generation,” that is, humans can use any (visual) medium to influence generative models’ outputs.
>
> [1] Shihao et al. “Uni-ControlNet: All-in-One Control to Text-to-Image Diffusion Models.” *NeurIPS 2023*.
>
> [2] Lin, et al. “Ctrl-Adapter: An Efficient and Versatile Framework for Adapting Diverse Controls to Any Diffusion Model.” *arXiv*:2404.09967.

---

### Official Review · Reviewer_6fgc · 2024-07-12

**Soundness:** 3
**Presentation:** 3
**Contribution:** 2
**Rating:** 5
**Confidence:** 5

**Summary:**

This paper proposes a training-free framework (Ctrl-X)to control the structure and appearance when diffusion generation without any training. The method does not need much more inference time cost or GPU resource cost.  The insight is that diffusion feature maps capture rich spatial structure and high-level appearance from early diffusion steps sufficient for structure and appearance control without guidance.  The experiments demonstrates superior results compared to previous training-based and guidance-based baselines (e.g. ControlNet + IP-Adapter [4, 5] and FreeControl [2]) in terms of condition alignment, text-image alignment, and image quality.

**Strengths:**

1. The method is novel and the motivation is clear.
2. The writing is good and easy to follow.

**Weaknesses:**

The main concern to me is the performance of the proposed method.
1. The authors do not provide any User Study results. They only show the quantitative results evaluated by DINO Self-sim and DINO-CLS, which are not widely used. In supp, they report the results in CLIP score, LPIPS in Table 3. From the result shown in Table 3, the proposed method gets a bad performance compared to others. For example, Ctrl-X gets the worst Self-sim performance.
2. The qualitative comparison is also unsatisfactory. For example, in Figure 5, I observe that ControlNet+ IP-Adapter gets a better result than Ctrl-X, e.g, the 3rd rows.

**Questions:**

See the weaknesses.

**Limitations:**

Yes.

---

> ### Author Rebuttal · Authors · 2024-08-06
>
> Thank you for your time and feedback on our work. We address your suggestions and concerns below. **You can find the referred figures in the PDF attached with the “global” response. Tables can similarly be found in the main body of the “global” response.**
>
> **Performance of Ctrl-X, qualitative evaluation.**
> Thanks for your suggestion of conducting a user study to prove the superior performance of Ctrl-X. We report the user evaluation in Table 1. We randomly selected 15 sample pairs from our dataset and then assign each sample pair to 7 methods: Splicing ViT Feature, Uni-ControlNet, ControlNet + IP-Adapter, T2I-Adapter + IP-Adapter, Cross-Image Attention, FreeControl, and Ctrl-X. We invited 10 users to evaluate pairs of results, each consisting of our method, Ctrl-X, and a baseline method. For each comparison, users assessed 15 pairs between Ctrl-X and each baseline, based on four criteria: “the quality of displayed images,” “the fidelity to the structure reference,” “the fidelity to the appearance reference,” and “overall fidelity to both structure and appearance reference.” We collected 150 comparison results for between Ctrl-X and each individual baseline method. We reported the human preference rate, which indicates the percentage of times participants preferred our results over the baselines. **The user study demonstrates that Ctrl-X outperforms training-free baselines and has a competitive performance compared to training-based baselines.**
>
> Moreover, we have included more qualitative Ctrl-X experiments on more challenging conditions (Figure 1), higher-level conditions (Figure 3), appearance- and structure-only control (Figures 4 and 5) in the PDF attached in the “global” response.
>
> **Performance of Ctrl-X, quantitative evaluation.**
> We respectfully disagree that DINO self-sim score is “not widely used.” The DINO self-sim score has been employed by previous works (InstructPix2Pix [1], Pix2Pix-Zero [2], Plug-and-Play [3], FreeControl [4]) to evaluate the similarity between the global structures of two images.
>
> Additionally, we do not think that “Ctrl-X has a bad performance.” Ctrl-X consistently achieves high fidelity to both structural and appearance references, and it performs better in terms of DINO self-sim and DINO-I scores compared to ControlNet + IP-Adapter and Cross-Image Attention. There is, in fact, a **trade-off** between structure consistency (DINO self-sim) and appearance similarity (DINO-I), as these are competing metrics—increasing structure preservation corresponds to worse appearance similarity, as shown in Figure 2 where we ablate Ctrl-X structure and appearance schedules. Single metrics are not representative of overall method performance, which is why we survey overall fidelity in our user study (Table 1), where Ctrl-X achieved the best overall fidelity. We will add examples to illustrate this trade-off in the camera-ready version.
>
> For the evaluation of appearance similarity, we re-evaluated all generated samples using the DINO-I score, which has been employed by appearance customization works such as DreamBooth [5], P+ [6], and Break-A-Scene [7]. The results are reported in Table 2. The DINO-I score computes the cosine similarity between the DINO [CLS] embeddings of two images, while our original DINO-CLS directly computes the mean square error of these embeddings. This new evaluation metric further demonstrates our promising performance compared to both training and training-free baselines.
>
> [1] Brooks et al. “InstructPix2Pix: Learning to Follow Image Editing Instructions.” *CVPR 2023*.
>
> [2] Parmar et al. “Zero-shot Image-to-Image Translation.” *SIGGRAPH 2023*.
>
> [3] Tumanyan et al. “Plug-and-Play Diffusion Features for Text-Driven Image-to-Image Translation.” *CVPR 2023*.
>
> [4] Sicheng et al. “FreeControl: Training-Free Spatial Control of Any Text-to-Image Diffusion Models with Any Condition.” *CVPR 2024*.
>
> [5] Ruiz et al. “DreamBooth: Fine Tuning Text-to-Image Diffusion Models for Subject-Driven Generation.” *CVPR 2023*.
>
> [6] Voynov et al. “P+: Extended Textual Conditioning in Text-to-Image Generation.” *arXiv*:2303.09522.
>
> [7] Avrahami et al. “Break-A-Scene: Extracting Multiple Concepts from a Single Image.” *SIGGRAPH Asia 2023*.

---

> > ### Comment · Reviewer_6fgc · 2024-08-13
> > **Thank you for the response**
> >
> > Thanks for the detailed response. Most of my concerns have been well-addressed. I have raised my rating from 4 to 5.

---

### Official Review · Reviewer_DRHE · 2024-07-12

**Soundness:** 3
**Presentation:** 3
**Contribution:** 3
**Rating:** 6
**Confidence:** 4

**Summary:**

This paper introduces a method for controllable generation using diffusion models. The approach is designed as a training free technique for 1) structure/layout controlled generation (like e.g. controlNet) and 2) appearance transfer. The approach leverages manipulation of attention mechanisms and information transfer from reference images to the generative image. Authors evaluate their methods in multiple settings, and compare to state of the art related work, showing that their approach can achieve similar performance to more expensive alternatives.

**Strengths:**

The paper is, for the most part, well written and well grounded in the related literature. Relevant related works are well references, and works from which ideas are borrowed are acknowledged.

The main benefit of the work is its fully training-free nature, which can offer more flexibility when generating images with different types of controls. The proposed methodology is relatively simple, leaving room for future improvements. Another benefit is the fact that the method can do both appearance and structure control, while related works often focus on a single one.

Experiments show promising results, with a performance similar to FreeControl, but without inference time optimization steps. Some ablations experiments are provided, in an effort to analyse the different  components of the model.

**Weaknesses:**

The main limitation of the work is the limited methodological novelty. The tools employed are not very novel: the structure transfer simply uses the method proposed in [34], while attention map manipulation is very commonly used for structure control (e.g. for editing methods such as prompt-to-prompt).

Another noticeable limitation, highlighted in Figure 9a, is the lack of flexibility with regards to using appearance OR structure control. Results show that appearance control is required when performing structure control, requiring to generate a separate appearance image. This can increase generation cost, and reduces control over the content of the image, as the appearance image is simply controlled by a prompt.

While the training free nature of the approach allows to use different types of structure images, the approach seems limited to control types with exact edge definition. Higher-level constraints like pose or bounding boxes do not appear to be an option with this type of method.

While the experiments on video generation are interesting, I would recommend that the authors focus more space on structure-only generation (as in the appendix) and expand the ablation and limitation experiments. For example, studying the impact on the quality of generated appearance image for structure-only generation would be beneficial.

**Questions:**

-Can the methodology handle pose or bounding boxes types of controls? If not, are there modifications that can be done to achieve this?
-Can the method handle images with more than one subject? All object centric experiments (structure + appearance) show generation with a single object, often at the center of the image. How does the approach perform with more complex images ?
-In certain settings, conditions appear to be too strong and can affect image quality (e.g. figure 11, dog image). The benefit of guidance/inference is that one can control the influence of a structure image, offering more flexibility. Is there a way to adjust the influence of a structure image in this approach?

**Limitations:**

One experiment in figure 9 shows one limitation of the proposed approach. However, as pointed out in above sections, there are several additional points that should be discussed (flexibility, single subject, loose controls, etc). For some of these, additional experiments could have allowed to understand the behaviour of the methodology more clearly.
Broader impact is adequately discussed.

---

> ### Author Rebuttal · Authors · 2024-08-06
>
> Thank you very much for the in-depth reading of our paper and the helpful comments. Our responses are listed below. **You can find the referred figures in the PDF attached with the “global” response. Tables can similarly be found in the main body of the “global” response.**
>
> **Model extension to multiple object generation.**
> Great suggestion! We conduct an additional experiment that involves multiple subjects in both the structure and appearance images, as seen in Figure 1. We tested Ctrl-X and ControlNet + IP-Adapter with two objects (house and tree) and three objects (dog, plant, and chair). Ctrl-X captures strong semantic correspondence between different objects and achieves balanced structure and appearance alignment. On the contrary, the training-based baseline often fails to maintain the structure and/or transfer the subjects’ appearance.
>
> **Adjust the influence of structure images.**
> Thanks for the suggestion! We present a new experiment that ablates different combinations of appearance and structure control schedules in Figure 2. Doing so changes the influence of the structure and appearance images on the output, making cross-class structure-appearance pairs (e.g., horse normal map with puppy appearance) look more realistic.
>
> **Limited to lower-level control types.**
> We present our experiments with new higher-level condition types: bounding boxes and human pose skeletons in Figure 3. Ctrl-X can handle these sparser and higher-level conditions by decreasing the structure schedule, making our method applicable to other higher-level control types, too.
>
> **Limited methodological novelty.**
> Indeed, Ctrl-X is inspired by previous literature on training-free structure control and appearance customization. However, we suggest the reviewer consider how our work advances guidance-free controllable text-to-image generation.
>
> Existing guidance-based methods have limited structure condition types and/or only handles either structure control or appearance control. Also, they require backpropagation through T2I/T2V which are only getting increasingly larger. In comparison, Ctrl-X enables fully disentangled control of structure and appearance while being fast and cheap in terms of inference time and GPU memory usage (Table 3). Moreover, whereas guidance-based methods are sensitive to guidance weights for each score function, Ctrl-X is more robust to its control schedules—in fact, the schedules can be varied to change the influence of the structure and appearance images (Figure 2).
>
> Compared to the training-based [34], where the query and key for attention come from trained embeddings, Ctrl-X directly uses the query and key within Stable Diffusion’s self-attention layers by exploiting the spatial correspondence property of self-attention observed by us and also prior training-free works. Thus, our method re-frames appearance control as a local style transfer task achieved by spatially-aware normalization of intermediate features. These techniques allow Ctrl-X to achieve multi-subject generation (Figure 1) and higher-level conditions like bounding box and pose (Figure 3) in an all-in-one method which none of our baselines can achieve.
>
> We believe our training-free and guidance-free method's disentanglement of structure control and appearance control, along with its flexibility and robustness, is novel, and we hope our method's utilization of attention layers is useful to the community—especially as visual generation moves towards transformer-based architectures.
>
> **Lack of flexibility with regards to using appearance OR structure control.**
> Thanks for the observation. Ctrl-X is specifically designed to solve the combined controllable generation by disentangling control from given structure and appearance images. Our baselines show that achieving a good balance between structure alignment and appearance transfer is often difficult, training-free or not. Thus, Ctrl-X aims to provide a solution for that.
>
> However, Ctrl-X *can* achieve appearance-only control by simply dropping structure control (and thus not needing to generate a structure image), as shown in Figure 4, displaying better appearance alignment for both the subject and background than the training-based IP-Adapter.
>
> Indeed, for structure-only control Ctrl-X needs to generate an appearance image, but Table 3 shows that the additional inference latency cost is not high and the peak GPU memory usage is in fact lower than ControlNet + IP-Adapter and T2I-Adapter + IP-Adapter as there are no additional modules. Plus, with multi-subject generation, we believe control over the content of the image can be achieved by simply providing an appearance image.
>
> **Impact of generated appearance image quality on structure-only generation.**
> We present structure-only generation samples alongside their jointly generated appearance images (equivalent to vanilla SDXL generation) in Figure 5. There is minimal quality difference between the generated appearance images and appearance-transferred output images, indicating that Ctrl-X’s appearance transfer does not greatly impact image quality. Thus, the output quality of Ctrl-X only grows as its base model’s quality improves.

---

### Official Review · Reviewer_tSTQ · 2024-07-14

**Soundness:** 4
**Presentation:** 4
**Contribution:** 4
**Rating:** 7
**Confidence:** 4

**Summary:**

This article presents Ctrl-X, a simple method for T2I diffusion models to control structure and appearance without additional training or guidance. Specifically, it uses feature injection and spatially-aware normalization in the attention laters to align the given structure and appearance. Doing so, Ctrl-X achieves training-free and guidance-free generation in both image and even videos. The effectiveness of Ctrl-X is demonstrated through experimental results on the collected benchmark, underscoring the model's capability.

**Strengths:**

Strengths

1. The technical elaboration of the proposed method is clear.
2. The motivation for the proposed method is straightforward. The insight of the paper is practical. Overall, I appreciate the high-level idea of this paper.
3. The evaluations conducted on the provided benchmarks provide evidence of the effectiveness of the proposed methods. However, there are some concerns regarding the experimental results, which will be further discussed in the weaknesses section.

**Weaknesses:**

Weaknesses

1. Model extension
I am curious whether this method can do multiple object generation. For example, given two object sketches/cannies in one picture, and two object appearances in another picture, can this framework automatically match the most suitable appearance and structure alignment for the two objects, thereby generating the picture?

2. Comparisons on latency
Since the paper claims that  other training-free models significantly increase computing time and require more GPU memory. It is suggested to add another table on comparing computing resources and inference latency.

**Questions:**

Shown as above

**Limitations:**

The paper has included this part in the conclusion section and checklist.

---

> ### Author Rebuttal · Authors · 2024-08-06
>
> Thank you for your positive feedback! We address your questions/concerns below. **You can find the referred figures in the PDF attached with the “global” response. Tables can similarly be found in the main body of the “global” response.**
>
> **Model extension to multiple-subject generation.**
> Great suggestion! We conduct an additional experiment that involves multiple subjects in both the structure and appearance images, as seen in Figure 1. We tested Ctrl-X and ControlNet + IP-Adapter with two objects (house and tree) and three objects (dog, plant, and chair). Ctrl-X captures strong semantic correspondence between different objects and achieves balanced structure and appearance alignment. On the contrary, the training-based baseline often fails to maintain the structure and/or transfer the subjects’ appearance.
>
> **Report peak GPU memory and inference latency.**
> Thanks for your suggestion. We report the inference latency and peak GPU memory usage in Table 3, re-tested on a single NVIDIA H100 GPU for a fair comparison. Ctrl-X (SDXL) is slightly slower than training-based baselines yet is significantly faster than training-free baselines and Splicing ViT Features. Moreover, Ctrl-X has lower peak GPU memory usage than SDXL training-based methods and significantly lower memory than SDXL training-free methods. (We note that Uni-ControlNet and Cross-Image attention uses the base model SD v1.5, which is ~4–5x faster and uses ~3x more memory compared to SDXL. Splicing ViT Features also trains its own much smaller custom model.)

---

### Author Rebuttal · Authors · 2024-08-06

We thank all the reviewers for their time and extensive reading of our paper. We are grateful that reviews find our paper “clear” (tSTQ, DRHE, 6fgc), our method effective (tSTQ, YZfZ), and our experiments promising (DRHE).

Responses to individual reviewers are addressed below each review. **Any referenced figures can be found in the attached one-page PDF on the “global” author rebuttal here** (which contain additional experiments and samples). **Any referenced tables are included below.** Please let us know if you have any additional questions or concerns!

---

For all quantitative and qualitative experiments in the below tables, we use SDXL as the base model whenever possible (ControlNet + IP-Adapter, T2I-Adapter + IP-Adapter, FreeControl, Ctrl-X); otherwise, we use the method's implemented/trained base model (SD v1.5 for Uni-ControlNet and Cross-Image Attention, custom model for Splicing ViT Features).

**Table 1: User study.** Average user preference of result quality, structure fidelity, appearance fidelity, and overall fidelity. We follow the setting of the user study from DenseDiffusion [1], where the human preference percentage showcases how often the participants preferred Ctrl-X over the baselines below. Ctrl-X consistently outperforms training-free baselines and is competitive with training-based ones, especially overal fidelity, showcasing Ctrl-X's ability to balance structure and appearance control.

| Method | Result quality &uarr; | Structure fidelity &uarr; | Appearance fidelity &uarr; | Overall fidelity &uarr; |
| :--- | :---: | :---: | :---: | :---: |
| Splicing ViT Features | 95% | 87% | 56% | 78% |
| Uni-ControlNet | 86% | 17% | 96% | 74% |
| ControlNet + IP-Adapter | 46% | 61% | 41% | 50% |
| T2I-Adapter + IP-Adapter | 74% | 53% | 67% | 58% |
| Cross-Image Attention | 95% | 83% | 83% | 83% |
| FreeControl | 64% | 48% | 79% | 74% |
| Ctrl-X (ours) | - | - | - | - |

**Table 2: Updated quantitative evaluation.** Updated quantitative evaluation with Uni-ControlNet and DINO-I appearance metric instead of DINO [CLS]. We use DINO-I following DreamBooth [2], which is the cosine similarity between the DINO ViT [CLS] tokens of the appearance and output images, where higher indicates better appearance transfer. Note that Splicing ViT Features scores are extremely high because it is trained per-image-pair and optimizes the loss between DINO features directly, as mentioned in our paper.

| Method | Self-sim: Natural image &darr; | DINO-I: Natural image &uarr; | Self-sim: ControlNet-supported &darr; | DINO-I: ControlNet-supported &uarr; | Self-sim: New condition &darr; | DINO-I: New condition &uarr; |
| :--- | :---: | :---: | :---: | :---: | :---: | :---: |
| Splicing ViT Features | 0.030 | 0.907 | 0.043 | 0.864 | 0.037 | 0.866 |
| Uni-ControlNet | **0.038** | 0.555 | **0.074** | 0.574 | **0.053** | 0.506 |
| ControlNet + IP-Adapter | 0.068 | *0.656* | 0.136 | *0.686* | 0.139 | *0.667* |
| T2I-Adapter + IP-Adapter | *0.055* | 0.603 | 0.118 | 0.586 | 0.109 | 0.566 |
| Cross-Image Attention | 0.145 | 0.651 | 0.196 | 0.510 | 0.195 | 0.570 |
| FreeControl | 0.058 | 0.572 | *0.101* | 0.585 | *0.089* | 0.567 |
| **Ctrl-X (ours)** | 0.057 | **0.686** | 0.121 | **0.698** | 0.109 | **0.676** |

**Table 3: Timing and computing resources.** Preprocessing time, inference time, and peak GPU memory of all methods. We re-test timing and add peak GPU memory usage of all methods on a single NVIDIA H100 GPU. Preprocessing time refers to method portions that are not the final sampling steps (e.g., feature extraction, inversion, etc.). Ctrl-X is slightly slower than training-based baselines yet significantly faster than training-free baselines and Splicing ViT Features. Moreover, Ctrl-X has lower peak GPU memory usage than SDXL training-based methods and significantly lower memory than SDXL training-free methods. (We note that Uni-ControlNet and Cross-Image attention uses the base model SD v1.5, which is ~4&ndash;5x faster and uses ~3x more memory compared to SDXL. Splicing ViT Features also trains its own much smaller custom model.)

| Method | Preprocessing time (s) | Inference latency (s) | Total time (s) | Peak GPU memory usage (GB) |
| :--- | :---: | :---: | :---: | :---: |
| Splicing ViT Features | 0.00 | 1557.09 | 1557.09 | 3.95 |
| Uni-ControlNet | 0.00 | 6.96 | 6.96 | 7.36 |
| ControlNet + IP-Adapter | 0.00 | 6.21 | 6.21 | 18.09 |
| T2I-Adapter + IP-Adapter | 0.00 | 4.37 | 4.37 | 13.28 |
| Cross-Image Attention | 18.33 | 24.47 | 42.80 | 8.85 |
| FreeControl | 239.36 | 139.53 | 378.89 | 44.34 |
| **Ctrl-X (ours)** | 0.00 | 10.91 | 10.91 | 11.51 |

[1] Kim et al., Dense Text-to-Image Generation with Attention Modulation, *ICCV 2023*.

[2] Ruiz et al. “DreamBooth: Fine Tuning Text-to-Image Diffusion Models for Subject-Driven Generation.” *CVPR 2023*.

---

### Author Response · Authors · 2024-08-11

We appreciate the reviewers for their thorough reading of our paper, and we look forward to engaging in detailed discussions and addressing the questions you have raised. Preparing our response may take some time, so we hope to begin our communication promptly.

---

### Decision · Program_Chairs · 2024-09-25

**Decision:**

Accept (poster)

**Comment:**

Reviewers either recommend acceptance or leaning towards acceptance. Reviewers appreciated the practically useful and training-free technique for structure and appearance control in diffusion models. Reviewers raised some concerns related to some missing analysis and discussion and several of which are addressed in the rebuttal. Reviewers did raise valid concerns and authors are encouraged to do the best of their abilities to address all the concerns in the final version.